# Combinatorial optimization of mRNA structure, stability, and translation for RNA-based therapeutics

Kathrin Leppek [1,11], Gun Woo Byeon[1,11], Wipapat Kladwang[2,11], Hannah K. Wayment-Steele[3,11], Craig H. Kerr[1,11], Adele F. Xu [1], Do Soon Kim[2], Ved V. Topkar[4], Christian Choe [5], Daphna Rothschild[1], Gerald C. Tiu[1], Roger Wellington-Oguri [6], Kotaro Fujii[1], Eesha Sharma[2], Andrew M. Watkins[2], John J. Nicol[6], Jonathan Romano [6,7], Bojan Tunguz[2,8], Fernando Diaz[9], Hui Cai[9], Pengbo Guo [9], Jiewei Wu[9], Fanyu Meng[9], Shuai Shi[9], Eterna Participants[6], Philip R. Dormitzer [9,10], Alicia Solórzano [9], Maria Barna [1✉] & Rhiju Das [2,4,6✉]

Therapeutic mRNAs and vaccines are being developed for a broad range of human diseases, including COVID-19. However, their optimization is hindered by mRNA instability and inefficient protein expression. Here, we describe design principles that overcome these barriers. We develop an RNA sequencing-based platform called PERSIST-seq to systematically delineate in-cell mRNA stability, ribosome load, as well as in-solution stability of a library of diverse mRNAs. We find that, surprisingly, in-cell stability is a greater driver of protein output than high ribosome load. We further introduce a method called In-line-seq, applied to thousands of diverse RNAs, that reveals sequence and structure-based rules for mitigating hydrolytic degradation. Our findings show that highly structured "superfolder" mRNAs can be designed to improve both stability and expression with further enhancement through pseudouridine nucleoside modification. Together, our study demonstrates simultaneous improvement of mRNA stability and protein expression and provides a computational-experimental platform for the enhancement of mRNA medicines.

[1] Department of Genetics, Stanford University, Stanford, CA 94305, USA. [2] Department of Biochemistry, Stanford University, Stanford, CA 94305, USA. [3] Department of Chemistry, Stanford University, Stanford, CA 94305, USA. [4] Program in Biophysics, Stanford University, Stanford, CA 94305, USA. [5] Department of Bioengineering, Stanford University, Stanford, CA 94305, USA. [6] Eterna Massive Open Laboratory, Stanford University, Stanford, CA 94305, USA. [7] Department of Computer Science and Engineering, State University of New York at Buffalo, Buffalo, New York 14260, USA. [8] NVIDIA Corporation, 2788 San Tomas Expy, Santa Clara, CA 95051, USA. [9] Pfizer Vaccine Research and Development, Pearl River, NY, USA. [10] Present address: GlaxoSmithKline, 1000 Winter St., Waltham, MA 02453, USA. [11] These authors contributed equally: Kathrin Leppek, Gun Woo Byeon, Wipapat Kladwang, Hannah K. Wayment-Steele, Craig H. Kerr. ✉email: mbarna@stanford.edu; rhiju@stanford.edu

Messenger RNA (mRNA) therapeutics hold the potential to transform modern medicine by providing a gene therapy platform with the capacity for rapid development and wide-scale deployment. Compared to recombinant proteins expressed in mammalian cell lines, manufacturing of mRNA is faster and more flexible because mRNA can be easily produced by in vitro transcription. Over the past decade, technological discoveries in the areas of mRNA modifications and delivery systems have rapidly advanced basic and clinical research in mRNA vaccines[1–3]. However, technical obstacles facing mRNA therapeutics are also apparent. For example, mRNA vaccines still suffer from decreased efficacy due to poor RNA stability in solution and in vivo and to limited expression of the payload mRNA; these are all pivotal issues that need to be carefully optimized for preclinical and clinical applications[3–6].

The development of mRNAs redesigned for increased stability and expression could maximize the impact of producing, delivering, and administering therapeutic mRNAs. However, achieving such designs is hindered by a poor understanding of how the sequence and structure of an mRNA influence its expression and stability, both in solution and in cells. For example, it has typically been assumed that mRNAs with more stable secondary structure might have increased in-solution stability but would have lower in-cell protein output due to the increased difficulty of the cellular translation machinery to process through RNA structure[7], but this has not been tested and some recent results suggest that there might not be such a tradeoff[8,9]. Thus, testing the expression efficiency of mRNAs that are predicted to have highly structured coding regions, which we term "superfolder" mRNAs[7], would be advantageous for the overall performance of therapeutic RNA. Our poor understanding of these design rules is due in part to the historical difficulty of rapidly synthesizing full-length mRNAs with different untranslated regions (UTRs) and coding sequences (CDSs) which would enable high-throughput experimental approaches comparing their stability and expression.

Here, we overcome these technical hurdles and characterize hundreds of full-length reporter constructs that encode mRNA sequences with a variety of UTRs and CDSs. We present a massively parallel reporter assay termed Pooled Evaluation of mRNA in-solution Stability, and In-cell Stability and Translation RNA-seq (PERSIST-seq), which enables systematic determination of the effects of UTR, codon sequence and RNA structure on mRNA translation in human cells and on mRNA stability, both in cells and in solution. This represents the first screen of hundreds of mRNAs redesigned across their entire length. We further leverage the unique ability of the Eterna[10] community, an online citizen science platform that enables participants to collectively solve RNA design puzzles, to devise solutions with high diversity in sequence and predicted structure. We integrate our datasets to develop a model that accurately predicts protein output for a given mRNA based on its ribosome load and in-cell stability. This model enables the identification of the best mRNA designs in our screen without the need to individually test all mRNAs for protein output. With the further aim of understanding the impact of mRNA structure on in-solution stability, we developed a high-throughput method termed In-line-seq to measure RNA degradation patterns arising from intrinsic in-line hydrolysis[11]. Nucleotide-resolution data on thousands of small, structured model RNAs from the Eterna platform revealed design parameters for reducing solution hydrolysis and enabled development of a regression model termed DegScore that enables in silico RNA sequence optimization for enhanced in-solution stability. Finally, we compare the effect of nucleoside modifications on mRNA performance, including results of pseudouridine (ψ) and its derivatives on in-solution stability. Ultimately, our findings

culminate in the fully automated design of highly structured "superfolder" mRNAs containing 5′ and 3′ UTR elements and structure-optimized CDS regions that simultaneously confer high stability in solution and high protein expression in cells. Together, the combination of select UTRs for expression, DegScore-optimized CDS structure, and ψ modification provides a general technology that can be applied to stabilize and increase protein expression of candidate mRNA therapeutics. We envision that these design rules and our combinatorial mRNA optimization platform will be widely applicable to rapidly engineer future mRNA therapeutics that simultaneously optimize stability and potency.

## Results

**A combinatorial library for systematic discovery of mRNA design rules.** In search of design rules for stable and high-expressing mRNAs, we aimed to characterize a large number of mRNA sequence designs with extensive variations in 5′ UTR, CDS, and 3′ UTR regions. We took advantage of recent accelerations in commercial gene synthesis and developed the massively parallel assay termed PERSIST-seq (Fig. 1a, Supplementary Fig. 1). In this method, mRNA variants can be assayed in parallel for translation efficiency in cells, for stability in cells, and for stability in solution. Full-length in vitro transcription (IVT) DNA templates were obtained through commercial gene synthesis services (Twist, Genscript, Codex). Each template incorporated three additional features: (1) a shared T7 promoter sequence for performing IVT, (2) barcodes in the 3′ UTR to enable multiplexing via inexpensive, short-read sequencing, and (3) a constant region at the 3′ end that enabled pooled PCR and reverse transcription (RT) reactions (Supplementary Fig. 1a–c). This design allowed one-pot amplification and analysis of the library using common flanking sequences. The whole library was in vitro transcribed and modified (3′ polyA-tailing and 5′ m7G-capping) together. We have no indication for any bias in the efficiency of such RNA modifications for certain sequences or structures in the different mRNA when performed in a pool (Supplementary Fig. 1c). The mRNA library was then transfected into cells and quantified by barcode sequencing in a pool (Fig. 1b), enabling straightforward measurements of translation by polysome profiling and of mRNA degradation over time in cells or in solution.

Our mRNA library contained 233 different mRNA sequences in total (Supplementary Data 1). Hundred and twelve of these mRNAs contained varied 5′ and/or 3′ UTRs (Fig. 1a). While there have been many efforts to increase protein expression by attaching UTRs found in highly translated or stable mRNAs[12–14], previous work tested and characterized only a handful of candidates at a time, either with a focus on individual functional UTR elements[15] or on screening randomized short UTRs[16–19]. We decided to harness full-length naturally occurring UTRs to test for increased mRNA expression, including 5′ and 3′ UTR sequences from cellular and viral genomes, in our systematic analysis due to their potential to enhance or fine-tune mRNA translation or stability.

As examples of cellular sequences, we included short 5′ UTRs of cellular mRNAs corresponding to highly abundant proteins that have a high translation rate such as ribosomal proteins (RPs) (*RPS25*, *RPL31*, *RPL38*), or structural components such as *tubulin beta-2B chain* (*Tubb2b*) and *actin* (*ActB*), as well as *human collagen, type I, alpha 2* (*hCOL1A2*)[20]. We further included regulatory 5′ UTR elements such as the 5′ terminal oligopyrimidine (TOP) motif from *RPL18*, which promotes translational activation downstream of mTOR[21,22], as well as the *Hoxa9* P4 RNA stem-loop which functions as a translation enhancer[23]. 5′ UTRs previously identified in translation efficiency screens such

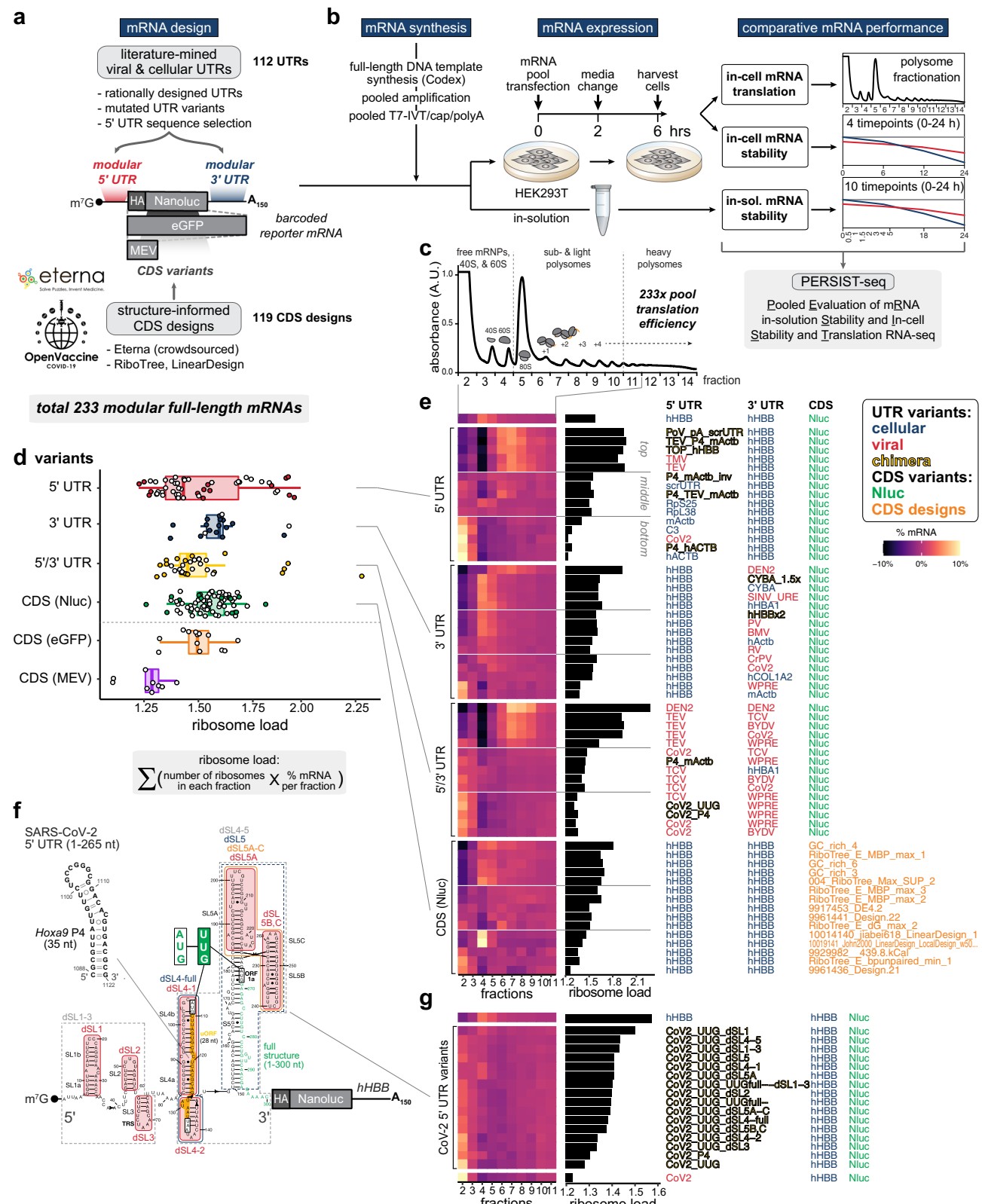

as *complement factor 3* (*C3*), *cytochrome P450 2E1* (*CYP2E1*), and *Apolipoprotein A-II* (*APOA2*)[13], as well as plant rubisco components (*RBCS3B*, *RBCS1A*)[24], were also included. For 3′ UTR regions (total 22; size range 60–597 nt), we employed known stabilizing RNA structures such as the MALAT1 non-coding RNA 3′-stem-loop structure that resembles an expression and nuclear retention element (ENE) and engages a downstream

A-rich tract in a triple helix structure[25,26], as well as known expression enhancing 3′ UTRs such as those from *human hemoglobin subunit alpha 1* (*HBA1*)[27] and *cytochrome B-245 alpha chain* (*CYBA*)[28,29].

Viruses have evolved a suite of compact regulatory elements for hijacking the host translation machinery to effectively promote translation and stability of their own mRNAs. For example,

**Fig. 1 PERSIST-seq overview and illustrative ribosome load insights. a** Overview of the mRNA optimization workflow. Literature mined and rationally designed 5′ and 3′ UTRs were combined with Eterna and algorithmically designed coding sequences. All sequences were then experimentally tested in parallel for in-solution and in-cell stability as well as ribosome load. The mRNA design included unique, 6–9 nt barcodes in the 3′ UTR for tag counting by short-read sequencing. **b** Experimental design for testing in-solution and in-cell stability and ribosome load in parallel. mRNAs were in vitro transcribed, 5′ capped, and polyadenylated in a pooled format before transfection into HEK293T cells or being subjected to in-solution degradation. Transfected cells were then harvested for sucrose gradient fractionation or in-cell degradation analysis. **c** Polysome trace from transfected HEK293T cells with 233-mRNA pool. **d** 5′ UTR variants display a higher variance in mean ribosome load per construct as determined from polysome sequencing. The formula for ribosome load is given. Box hinges: 25% quantile, median, 75% quantile, respectively, from left to right. Whiskers: lower or upper hinge ±1.5 x interquartile range. **e** Heatmaps from polysome profiles of mRNA designs selected from the top, middle, and bottom five mRNAs (by ribosome load) from each design category. **f** Secondary structure model of the SARS-CoV-2 5′ UTR. Introduced mutations and substitutions are highlighted. **g** Heatmaps of SARS-CoV-2 5′ UTR variants' polysome profiles sorted by ribosome load.

internal ribosome entry sites (IRESs) can recruit ribosomes to initiate translation without the need for the full repertoire of eukaryotic initiation factors, whereas other elements in the 5′ and 3′ UTRs of viruses encode structures that facilitate long-range RNA-RNA interactions to enhance protein expression or mRNA stability. Therefore, several UTRs originating from viral genomes were included: the 5′ and 3′ UTR elements of the SARS-CoV-2 RNA genome[30,31], along with variants described below; the dengue virus (DEN2) 5′ and 3′ UTRs, which are thought to enhance viral protein expression[32,33]; 5′ and 3′ UTR elements from various tombusviruses (e.g., turnip crinkle virus (TCV)) which encode 3′ cap-independent translational enhancer RNA structures that recruit the translational machinery[34,35]; tobacco mosaic virus[36] (TMV) and tobacco etch virus[37] (TEV) 5′ leader sequences; a poxvirus poly(A) leader sequence that is proposed to facilitate translation[38]; and the 3′ UTRs of Sindbis virus (SINV) and the rabies virus glycoprotein, which increases viral RNA stability through recruitment of host proteins[39–41].

As our main reference, we chose the 5′ and 3′ UTRs from human hemoglobin subunit beta (hHBB), which is one of the most efficiently expressed mammalian mRNAs and is commonly used in investigations of mRNA translation and stability[42,43]. Non-hHBB UTRs are referred to here as "UTR variants." To test these UTR variants, all reporter mRNAs encode the Nanoluc luciferase (Nluc) open reading frame (ORF) as its CDS region[44,45]. We decided to use Nluc because its short ORF of 621 nt allowed for synthesis and pooled amplification of full-length DNA templates with UTRs of up to 600 nt attached at each end. Employing Nluc further enabled precise quantitative readout for comparing translation efficiencies in follow-up experiments on individual mRNAs through ratiometric measurements including a firefly luciferase (Fluc) mRNA spike-in control in transfection experiments integrated with luciferase luminescence assays[23]. Additional mRNAs encoding for enhanced green fluorescent protein (eGFP) and a shorter candidate multi-epitope vaccine (MEV)[7] were included as controls in some experiments, further discussed below.

To test the impact of CDS sequence and predicted CDS structure on mRNA stability and translation, we sought to maximize the diversity of CDS sequences and structures for model protein targets. Consequently, we asked participants from the Eterna massive open laboratory[10] to design CDSs encoding a variety of model mRNAs, without specific optimization metrics, in a series of challenges (Fig. 1a). These puzzles included design challenges for eGFP, MEV, and Nluc ('OpenVaccine: Design of eGFP and epitope mRNA molecules' and 'OpenVaccine: Lightning Round Design + Vote of Nanoluciferase'). Later rounds of design challenges included degradation-specific metrics (AUP and DegScore, discussed later) within the game interface to guide optimization. We also included CDSs using several algorithmic approaches. First, we included sequences designed using commercially available algorithms to optimize codon adaptation

index (CAI)[46]. Second, we designed sequences using a "GC-rich" approach, in which each codon is stochastically sampled from codons highest in GC content, based on a strategy developed by CureVac[9]. Third, we included CDSs designed using the LinearDesign algorithm[47], which returns a deterministic minimal free energy solution that is weighted by codon optimality. Finally, we used the Ribotree Monte Carlo tree search method to optimize AUP for eGFP and to compare them to eGFP designs developed by Moderna[7,8]. These design methods yielded a total of 121 CDS variants in the library (Fig. 1a, Supplementary Data 1). To ensure useful cross comparison, hHBB UTRs were used for each of these CDS variants. Together, all candidates from these diverse sources of UTR and CDS sequences were combined into a single library of 233 mRNA constructs.

**High dynamic range of translation driven by UTRs.** To assess translation efficiencies, PERSIST-seq uses transfection of mRNA pools into human cells (here HEK293T). The cell lysate then undergoes sucrose gradient fractionation, which separates mRNAs into actively translating and non-translating fractions that are analyzed by RT-PCR of barcode regions and Illumina sequencing. Actively translating mRNAs have a higher number of ribosomes associated with them and are found in polysomal fractions whereas non-translating or poorly translating mRNAs are present in the free mRNA fraction or are associated with 40S ribosomal subunits (Fig. 1c). After initial studies confirming differences in polysome loading of a highly translated endogenous mRNA, human ActB, with that of a transfected control mRNA that has scrambled short UTR sequences[44] (Supplementary Fig. 1d), we carried out PERSIST-seq to examine the polysome profiles of diverse constructs in the 233x-mRNA library (Supplementary Fig. 1f). In parallel, we perform PERSIST-seq to compare the in-cell and in-solution stability of the mRNA designs in the same library over time (Supplementary Fig. 1e). Importantly, we are quantifying relative differences across polysome fractions or time points, from which we calculate ribosome load (Supplementary Fig. 1f), or fit decay curves for half-lives (Supplementary Fig. 1g), respectively. Thus, any construct-specific biases in RT efficiencies are regressed out (i.e., only the relative ratios of the counts between time points, not the absolute read counts, are relevant for interpreting stability). We observed a wide variation in mRNA distribution across the fractions (here expressed as ribosome load, defined as the weighted sum of mRNA proportions multiplied by the ribosome number in a fraction) (equation in Fig. 1d). The largest variation in ribosome load was observed for the 5′ UTR variants group (Fig. 1d). These data suggested a strong potential for using different 5′ UTRs to tune the translational efficiency of target mRNAs—more than for any other region.

We categorized the origins of UTR sequences described above as "cellular", "viral", and "chimera" (modular UTR combinations). Overall, the mRNA designs with highest ribosome load

were observed across 5′ UTRs of cellular as well as viral origins (Fig. 1e, Supplementary Data 1). These 5′ UTRs included: mouse *COL1A2*, *Hoxa9* P4, *Rpl18a* TOP, plant *RBCS1A*; the poxvirus poly(A) leader sequence fused to a scrambled 5′ UTR sequence and also the 5′ UTRs of plant viruses TEV and TMV. The dengue virus 5′ and 3′ UTRs both individually increase ribosome loading, and combining them into one mRNA resulted in an additive effect (Fig. 1e, Supplementary Data 1). All of these sequences had a higher ribosome load (1.7–2.3) than the *hHBB* 5′ UTR (1.57), thereby identifying potential UTR design strategies to boost mRNA translational efficiency. Moreover, the chimeric fusion of the *hHBB* 5′ UTR with elements such as the TEV or 5′ TOP sequence in the same 5′ UTR increased polysome loading. We also found that human beta-actin (*hACTB*) does not perform well in terms of ribosome loading but the mouse beta-actin (*mActb*) performs in the same range as the scrambled short 5′ UTR (*scrUTR*) or *hHBB*. Overall, our polysome screen successfully identified a wide range of 5′ UTR sequences that can be deployed to successfully promote translation within cells. Most surprising, in contrast to previous reports[48–50], we found 5′ UTRs that are highly structured, such as the dengue virus (DEN2), can support efficient translation initiation. Our present studies do not permit us to more directly compare the effect of structured 5′UTRs on translation beyond individual candidates.

We next asked if the highly structured 5′ UTRs used in our screen could be augmented to further improve ribosome loading. While the combined DEN2 3′ and 5′ UTRs are well loaded with ribosomes (Fig. 1e), the DEN2 5′ UTR alone did not perform as one of the top candidates 5′ UTRs. To this end, we chose the SARS-CoV-2 5′ UTR to further optimize as it is much more highly structured than DEN2 5′ UTR while still sustaining sufficient translation levels[31]. We performed a detailed mutagenesis analysis of the structured 5′ UTR[31,51–53] from SARS-CoV-2 genomic RNA[30,31] as an example (Fig. 1f). We first observed that mutation of the uORF in the 5′ UTR (AUG mutated to UUG; CoV2-UUG) resulted in a higher ribosome load than the wildtype 5′ UTR (Fig. 1f, g). Then, to systematically determine the impact of each stem-loop on ribosome load, we introduced partial and full truncations of stem-loops (SL) 1-5 (SL1-5) into CoV-2-UUG (Fig. 1f, Supplementary Data 2). In addition, we included larger truncations of combined deletions of adjacent stem-loops and introduced the *Hoxa9* P4 stem-loop, a 35 nt element that recruits 40S ribosomal subunits to enhance translation[23], in lieu of the uORF (Fig. 1f, g, Supplementary Data 2). Intriguingly, polysome profiling of the 5′ UTR SL mutant constructs revealed a wide range of overall improved ribosome load (Fig. 1g). Deleting the full 5′ half (dSL1-3) or the 3′ half (dSL4-5) of the 5′ UTR overall increased ribosome load. In particular, deletion of SL1 from the SARS-CoV-2 5′ UTR increased ribosome load from 1.23 (CoV-2) to 1.5 (CoV-2-UUG-dSL-1) which almost reaches the level of ribosome load of *hHBB* (1.57) (Fig. 1g). These results on the SARS-CoV-2 5′ UTR indicate that ribosome load can be finetuned through the modulation of distinct elements in structured viral 5′ UTRs, and that these effects can be read out through PERSIST-seq.

Inspired by our PERSIST-seq results, we sought to further understand how sequence and structure variation of 5′ UTRs might modulate translation efficiency through an unbiased selection from a complex sequence library (Supplementary Fig. 2, and see Supplementary Note 1 for details). We selected for highly translating transcripts by transfecting an mRNA reporter library containing randomized 5′ UTR sequences and harvesting mRNAs associated with heavy polysomes (Supplementary Fig. 2a). We further enriched these libraries for highly translating transcripts over five total rounds of selection and re-transfection of the heavily ribosome-loaded mRNAs from two independent starting

pools (Supplementary Fig. 2a–c). We then functionally assessed the protein output of the top 15 sequences by normalized read abundance (Supplementary Fig. 2d). We found the most impactful effects of sequence k-mers at the 5′ and 3′ end of the randomized 5′ UTR stretch (Supplementary Fig. 2e, Supplementary Data 3). These effects included significant depletion of 5′ UTRs that contain out-of-frame AUG start codons relative to the main ORF and enrichments of short stem-loop motifs promoting translation (Supplementary Fig. 2f). Interestingly, one-by-one tests of the 5′ UTRs selected to have high ribosome load gave lower total protein output than the starting sequence (Supplementary Fig. 2d; see Supplementary Note 1); this observation foreshadowed a tradeoff between ribosome load and mRNA stability that we dissect in greater detail below.

Beyond effects of varying sequence and structure of UTRs, we assessed how variations of the CDS might impact translation[3,8,46]. In particular, variation in the CDS may result in higher ribosome load due to enhanced overall translation (i.e., higher loading, faster elongation) or from increased dwell time of ribosomes (i.e., equal loading, slower elongation) on a given transcript as a result of RNA secondary structure. Interestingly, we observed less variance in ribosome loading among CDS variants than for UTR variants (Fig. 1d, e). The only clear effects on ribosome load were that MEV-encoding CDS variants displayed less ribosome load than Nluc- or eGFP-encoding CDS variants, as expected from their shorter ORFs. The ribosome load of CDS variants had no obvious correlation with the CAI, GC content, minimum free energy (MFE), the addition of signal peptides, or nonsynonymous mutations (Supplementary Data 1, Supplementary Fig. 3a). Specifically, for "GC_Rich" designs we observed a similar ribosome load to the reference Nluc sequence despite these designs having higher than average CAI values (Fig. 1e). It may be expected that these constructs would have a lower ribosome load as ribosomes are thought to translocate 'faster' on transcripts with more optimal codons[54]. Nevertheless, as these CDSs were designed to form extensive secondary structure rather than optimal CAI, it is thus possible that the highly structured CDS counteracts the codon optimality. In addition, we note that mRNAs designed to have highly structured CDSs by the LinearDesign algorithm exhibited polysome fraction profiles dominated by 80S monosomes (Fig. 1e), which is further discussed below. Taken together with our results above, these PERSIST-seq measurements indicate that structured CDS regions and a variety of 5′ UTR elements from cellular and viral origins can sustain or even improve in-cell translation efficiency compared to reference mRNA sequences.

## In-cell mRNA stability is a major predictor of total protein output

The total protein output from an mRNA depends not only on its in-cell translational rate but also on how long it remains intact inside cells. To assess the in-cell mRNA stability of the library of constructs in a pooled fashion, PERSIST-seq quantifies the fractions of mRNAs remaining at multiple timepoints following transfection of the library into cells. To ensure recovery of intact full-length mRNAs rather than their degraded fragments, PERSIST-seq uses a two-step protocol, first generating amplicons covering the entire CDS regions of mRNAs through reverse transcription-PCR (RT-PCR) and then using primers flanking just the barcode region for a second PCR before shortread Illumina sequencing to count intact mRNAs per time point (Supplementary Fig. 1e).

For our library of 233 mRNAs, PERSIST-seq gives a dynamic range of in-cell half-lives, ranging from <5 h to over 15 h (Fig. 2a). We had originally expected that the sub-library comprised of varying 3′ UTR sequences would give the most variation in in-cell

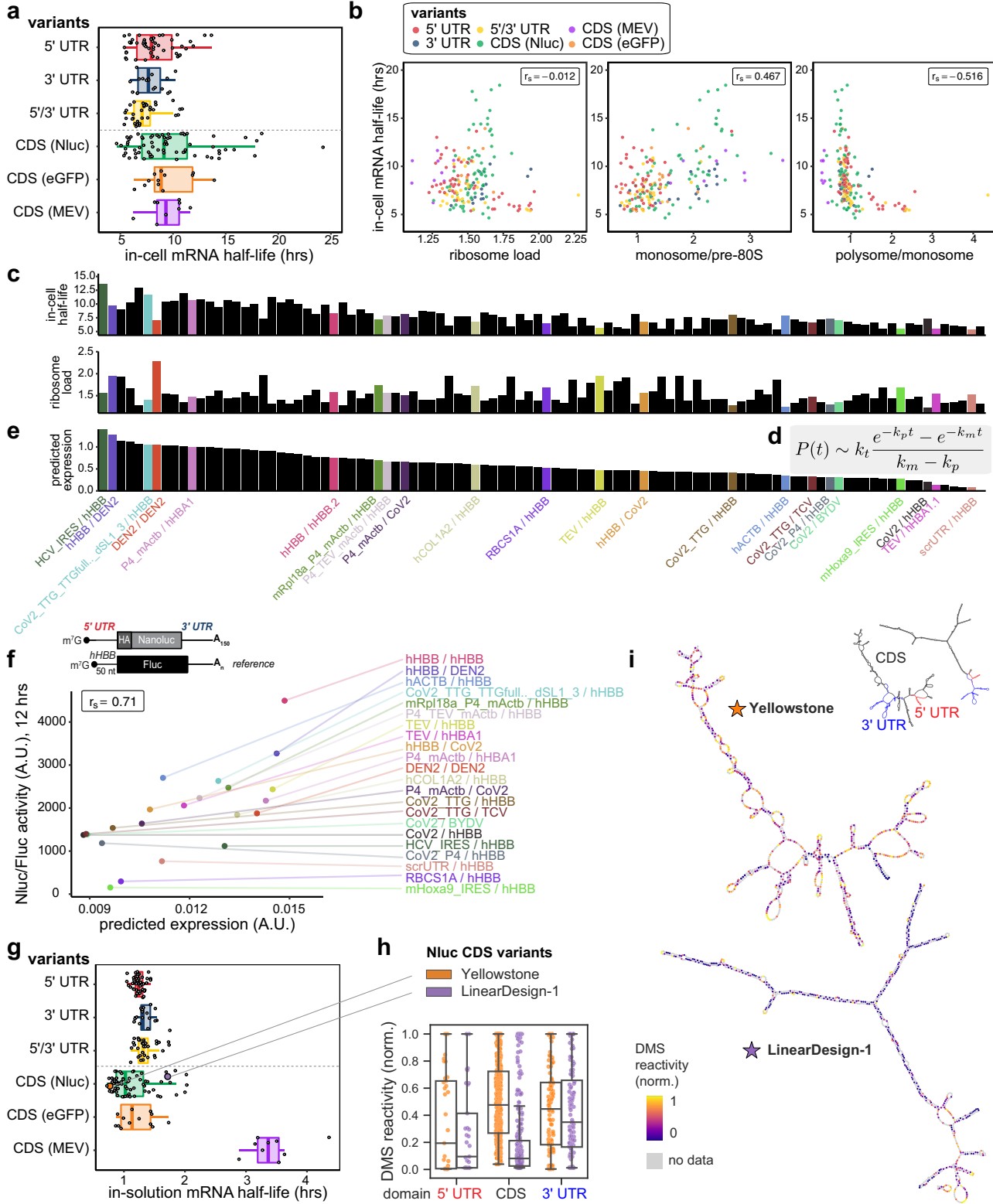

stability, since those mRNAs included diverse *cis*-regulatory elements that are known to recruit cytoplasmic factors to aid or prevent mRNA decay[55–58]. However, we instead observed the widest in-cell stability variation in the CDS and 5′ UTR variant groups (Fig. 2a). We also noted a non-linear relationship between ribosome load and mRNA stability, where there is a positive association between a moderate increase in ribosome load and mRNA stability while the opposite is true for the highest

ribosome load values (Fig. 2b, Supplementary Fig. 3). Examining this more closely, we find that metrics that separately look at polysome versus monosome loading better capture this trend: high polysome/monosome ratios predict lower stability ($r_S = -0.516$) while monosome/pre-80S ratios predict increased stability ($r_S = 0.467$). These findings identify an unexpected rule for the design of mRNA: increasing translation efficiency may counterintuitively decrease protein expression, when excessive

**Fig. 2 In-cell RNA stability drives downstream protein expression levels. a** In-cell half-life of each mRNA design in HEK293T cells. Box hinges: 25% quantile, median, 75% quantile, respectively, from left to right. Whiskers: lower or upper hinge ±1.5 x interquartile range. **b** Higher polysome load correlates with decreased in-cell half-life. Correlation between in-cell half-life and mean ribosome load across the entire profile (left), monosome-to-free subunit ratio (center), or polysome-to-monosome ratio (right). **c** In-cell half-life and mean ribosome load for individual mRNA designs with varying UTRs. **d** Kinetic model for predicting protein expression from mRNA half-life and ribosome load. $P(t)$ is protein quantity at time $t$; $k_t$ is translation rate; and $k_m$ and $k_p$ are rates of mRNA and protein decay, respectively. **e** Protein expression predicted using the kinetic model in (**d**) on the basis of mRNA half-life and ribosome load. Predicted protein expression of each UTR variant; note closer similarity to in-cell half-life data than to ribosome load in (**c**). **f** Correlation of predicted protein expression and Nluc/Fluc activity at 12 h in HEK293T cells. Predicted protein expression is normalized by mRNA length (corresponding to transfecting equal masses of each mRNA). **g** In-solution half-life of various mRNA design variants. mRNA lifetimes are strongly dependent on mRNA length and designed structures, revealed by time courses of mRNA degradation under accelerated aging conditions (10 mM $MgCl_2$, 50 mM Na-CHES, pH 10.0). Box hinges: 25% quantile, median, 75% quantile, respectively, from left to right. Whiskers: lower or upper hinge ±1.5 x interquartile range. **h** Nucleotide-resolution in vitro DMS mapping confirms large differences in structural accessibility between a highly structured JEV-HA-Nluc mRNA construct, "LinearDesign-1" and a highly unstructured construct "Yellowstone". The 5′ and 3′ UTRs (*hHBB*) were kept constant between designs. Each point represents normalized DMS reactivity from one nucleotide position of the RNA. Box plot represents median and 25th and 75th percentiles—interquartile range; IQR—and whiskers extend to maximum and minimum values. **i** Nucleotide DMS accessibility mapped onto structures from DMS-directed structure prediction.

polysome loading leads to decreased mRNA stability such that the total amount of protein produced over time actually becomes lower[16].

To explore the implications of this apparent tradeoff, we sought to integrate both translation (ribosome load) and in-cell stability (Fig. 2c) into a simple quantitative model to understand and predict their relative impact on integrated protein expression (Fig. 2d, e)[59]. To this end, we used differential equations to describe biochemical kinetics of mRNA translation and mRNA/protein decay assuming first-order rate of translation and exponential decay for mRNAs and for proteins (Fig. 2d).

This expression enables the ranking of expected protein levels across the UTR variants in our mRNA library, as $k_t$ and $k_m$ are related to PERSIST-seq measurements of ribosome load and in-cell half-lives, respectively, and the other parameters are known (Fig. 2e). Overall, predicted protein expression per mole of transfected mRNA is modeled to be mostly driven by the in-cell mRNA half-life (Fig. 2c, top panel), which more closely trends with predicted protein expression (Fig. 2e) than ribosome load (Fig. 2c, lower panel). To test this model, we predicted and measured, using the luciferase readout, the protein levels observed at a given time (here 12 h post-transfection) when an equal mass of mRNA is transfected for each construct. The predicted and measured luciferase activities showed a correlation of $r = 0.71$ (Fig. 2f), supporting the accuracy of our model. However, for short expression (early time points (6 h) after mRNA transfection) or for "above-threshold" protein half-lives, translation can become the dominant predictor. Indeed, the analysis of the correlations of ribosome load and in-cell mRNA half-life with protein expression at 6 and 12 h post transfection indicates that only at later time points (12 h), mRNA half-life is a driver and an approximate predictor of protein expression (Supplementary Fig. 4b, $r_s = 0.41$). Therefore, depending on the desired parameters (early burst of high protein expression or total protein output integrated over a longer time), the most optimal set of UTRs can vary. Taken together, our large-scale and unbiased pooled measurement of translation and mRNA stability provides a platform to test large numbers of different mRNA sequence designs for desired protein expression and has allowed us to infer that in-cell mRNA stability is the dominant predictor of total protein output (Fig. 2c, e).

**mRNA length and structure drive in-solution mRNA stability.** The degradation of RNA in solution is a major obstacle in the distribution of mRNA therapeutics to patients[6]. Thus, our final use of PERSIST-seq was to evaluate structure-based RNA design strategies to yield more stable RNAs in solution. Just as with the

in-cell stability measurements, we measured the fraction of mRNA CDS regions that remain intact after degradation, taking advantage of the same RT-PCR to select for intact mRNAs, followed by a PCR amplifying the short barcode regions in the 3′ UTR (Fig. 1a, Supplementary Fig. 1a, e). To mimic the high effective pH and positively charged environment that can arise in lipid nanoparticles, protamine, and other formulations for mRNA therapeutics[6], we used a high pH buffer containing magnesium ($Mg^{2+}$) to accelerate degradation (10 mM $MgCl_2$, 50 mM Na-CHES, pH 10.0, 24 °C); conditions without $Mg^{2+}$ or at lower pH lead to similar conclusions on relative stabilities across RNA variants (see below). The results for in-solution stability were very different from the results for in-cell stability across the mRNA library. For example, in cells, modulation of the UTR sequence produced large variation in in-cell stability (Fig. 2a), presumably through changes in recruitment of cellular machinery that affect mRNA decay. In contrast, in aqueous solution without such cellular factors, changing UTRs produced comparably little change in RNA stability to hydrolytic degradation of the CDS (Fig. 2g).

A greater than threefold change in in-solution half-lives was observed across CDS variants. The strongest prediction of previous theoretical modeling[7] was that length changes should drive the most variation in in-solution stability, and PERSIST-seq data across different CDS types confirmed the effect of CDS length on RNA stability (Fig. 2g). The shortest mRNAs in the pool, encoding a multi-epitope SARS-CoV-2 vaccine CDS (MEV), exhibited in-solution half-lives of 3.4 ± 0.6 h. The longest mRNAs, encoding eGFP, exhibited much shorter in-solution half-lives of 1.1 ± 0.08 h (Fig. 2g), as expected given the larger number of sites of potential hydrolysis. Indeed, the ratio of these half-lives, 3.0 ± 0.7, matched within error the inverse ratio of the lengths of the mRNA regions captured by RT-PCR (958 nt/250 nt = 3.8), supporting theoretical predictions of length effects.

The next largest source of variation in in-solution stability was driven by differences in mRNA structure. Within mRNAs encoding for a single protein (Nluc, eGFP, or MEV), the variance in in-solution half-lives was greater than for the variance across UTR variants (Fig. 2g), and these values correlated well with different metrics for predicted structure (see below). The largest spread of in-solution half-lives (2.8-fold) occurred in the CDS variants for Nluc mRNAs. We chose two Eterna-submitted solutions amongst these mRNAs with short and long half-lives for follow-up: 'Yellowstone', a design using codons that mimic the base frequencies (high A/C content) found in organisms found in the Yellowstone hot springs[60]; and 'LinearDesign-1', a design based on the LinearDesign mRNA structure optimization

server[47]. Chemical structure mapping showed that the long-lived LinearDesign-1 was significantly more highly structured than Yellowstone, as assessed by dimethyl sulfate (DMS) and selective 2′-hydroxyl acylation with primer extension (SHAPE) reactivities[61,62] (Fig. 2h, Supplementary Fig. 5, Supplementary Data 4) and structure models guided by these data (Fig. 2i). Overall, the global assessment of in-solution RNA degradation using PERSIST-Seq reveals the effects of RNA length and structure on RNA half-life in solution.

**Eterna-guided In-line-seq yields additional design principles and DegScore predictor.** The mRNA designs above varied in-solution stability based mainly on computational predictions of mRNA structure and the assumption that nucleotides that are not base paired in structure would be uniformly prone to hydrolytic degradation[7]. We hypothesized that we might further improve in-solution stability through a deeper understanding of any specific sequence and structure features that lead to enhanced or suppressed hydrolysis in such unpaired regions. For example, base identities and local structural features such as the size and symmetry of apical loops and internal loops may play roles in determining in-solution mRNA degradation[11,63,64].

To test such effects and to potentially discover unknown ones, we challenged Eterna participants to generate a large and diverse set of RNA molecules featuring designed secondary structure motifs in a special challenge ('OpenVaccine: Roll-your-own-structure', RYOS). Limiting the lengths of these molecules to 68 nucleotides and soliciting unique 3′ barcode hairpins enabled massively parallel synthesis and characterization of thousands of RNA molecules for their structure and degradation profiles (Fig. 3a). In particular, we obtained single-nucleotide-resolution measurements of 3030 RNA fragments using In-line-seq, a version of a low-bias ligation and reverse transcription protocol (MAP-seq)[65] adapted here for in-line hydrolysis profiles[66] (Fig. 3b). This is a massively parallel methodology and large-scale data set applying in-line probing to RNA. We compared degradation profiles from In-line-seq to profiles from constructs whose in-line degradation was probed one-by-one and read out with capillary electrophoresis, which confirmed excellent agreement between In-line-seq and low-throughput capillary electrophoresis (Supplementary Fig. 6). For analysis of sequence and structure motifs, sequences were filtered for low experimental noise. Then, we ensured that the structure predicted in ViennaRNA[67] matched the structure inferred through SHAPE mapping data collected at the same time (see Methods). These filters resulted in 2165 sequences and corresponding secondary structures. We matched the accelerated degradation conditions used for PERSIST-seq in-solution stability measurements, but also verified that calculated degradation rates without $Mg^{2+}$, at lower pH, and higher temperature gave strongly correlated results (Supplementary Fig. 7a). At a broad level, the data confirmed that RNA structure was a dominant predictor of in-line hydrolysis rate (compare, e.g., SHAPE to in-line data; Supplementary Fig. 7a), but a closer look revealed additional sequence and structure-dependent rules for in-line hydrolysis.

When analyzed across known secondary structure motifs, the data revealed that the RNA sequence in a given structure can dramatically affect degradation of the structure motif. For example, in the case of the most-sampled secondary structure of triloops, the in-line hydrolysis rates varied by up to 100-fold depending on sequence (Supplementary Fig. 7b). Furthermore, in many RNA loop types, it appeared that linkages that lead to a 3′ uridine were particularly prone to degradation (Fig. 3c, Supplementary Fig. 7c). This effect was reproduced in follow-up experiments by capillary electrophoresis (Supplementary Fig. 7d).

Thus, independent of the nucleotide identity 5′ of the U, this bond is a hotspot for in-line nucleophilic attack[11,66]. In addition, we noted rules for hydrolytic degradation that depended on the type of RNA structural loop in which a nucleotide appears. We visualized sequence dependence within triloops by aggregating all triloops by sequence position at each position in the triloop ($n > 5$). Each visualization in Fig. 3c represents the median degradation measure ($n > 5$). A particularly salient characteristic was suppressed hydrolysis in symmetric internal loops compared to asymmetric internal loops (Fig. 3c). To distill these observations into a predictive model, we trained a windowed ridge regression model called 'DegScore' based on these In-line-seq data (Fig. 3d; see Methods) which quantitatively captured features like the increased hydrolysis rates at linkages leading to 3′ U (Fig. 3d). The DegScore regression coefficients with the largest magnitude corresponded to the identity of the nucleotide 3′ of a linkage (Fig. 3d). Of these coefficients, G and C were the most favorable (least hydrolysis) to have 3′ of the linkage, followed by A, and a 3′ U was most detrimental to degradation, matching our prior observations.

To test the accuracy of the DegScore metric derived from In-line-seq data, we made predictions of in-solution half-lives for the mRNAs measured in the PERSIST-seq experiments, which were carried out independently (Fig. 1b, Supplementary Fig. 2g). For the Nluc CDS variants that showed the widest variance in in-solution half-lives, we observed a strong correlation of DegScore predictions to the in-solution half-lives (Spearman $R = -0.66$, $p < 0.0001$). Strikingly, the accuracy of DegScore outperformed the accuracy of two other metrics described before to parametrize RNA structure but which do not take into account sequence or structure-motif dependences of RNA hydrolysis: the free energy of the predicted MFE secondary structure, the metric used in several design algorithms including LinearDesign ($dG_{MFE}$; $R = -0.50$), and the predicted summed unpaired probability of the RNA structure ensemble[7] (SUP; $R = -0.62$) (Fig. 3e). The difference in correlation coefficients were not significant with $p < 0.05$ as evaluated by a two-sided significance test on dependent overlapping correlations[68] (see Methods); more experimental studies are likely needed to achieve significance for these and future metrics. Beyond the Nluc CDS variants, we confirmed that DegScore gave the highest accuracy in predicting in-solution stability when evaluated over all the measured mRNAs, including low and high structure eGFP mRNAs from Moderna researchers, Eterna, and Ribotree (Supplementary Table 1 and Supplementary Fig. 8a, b).

**Pseudouridine stabilizes RNA in solution.** Given that the linkages with 3′ U were particularly sensitive to degradation, we hypothesized that the base chemistry of U may be directly linked to the degradative capacity of this nucleoside and sought to test whether chemical alternatives to U might alleviate degradation. In particular, we focused on ψ and m[1]ψ, since these substitutions for U have been widely adopted for mRNA therapeutics and vaccines due to improved in-cell translation and to better control of innate immune responses[2,3,8,69] through avoidance of recognition by cellular toll-like receptors (TLR7 and TLR8)[70], RIG-I, and PKR[71–73]. While ψ and derivatives have been reported to stabilize mRNAs against decay in cells[69], the effects of these modifications on mRNA stability in solution have not been reported. We selected RNA sequences from the Eterna RYOS challenge (Fig. 3a) that were designed to contain U-rich loops or U-rich unstructured regions, resynthesized these RNAs with standard nucleotides or with ψ or m[1]ψ substituted for U, and measured their in-line degradation over time via capillary electrophoresis. We observed that substitution of U with either ψ or m[1]ψ led to

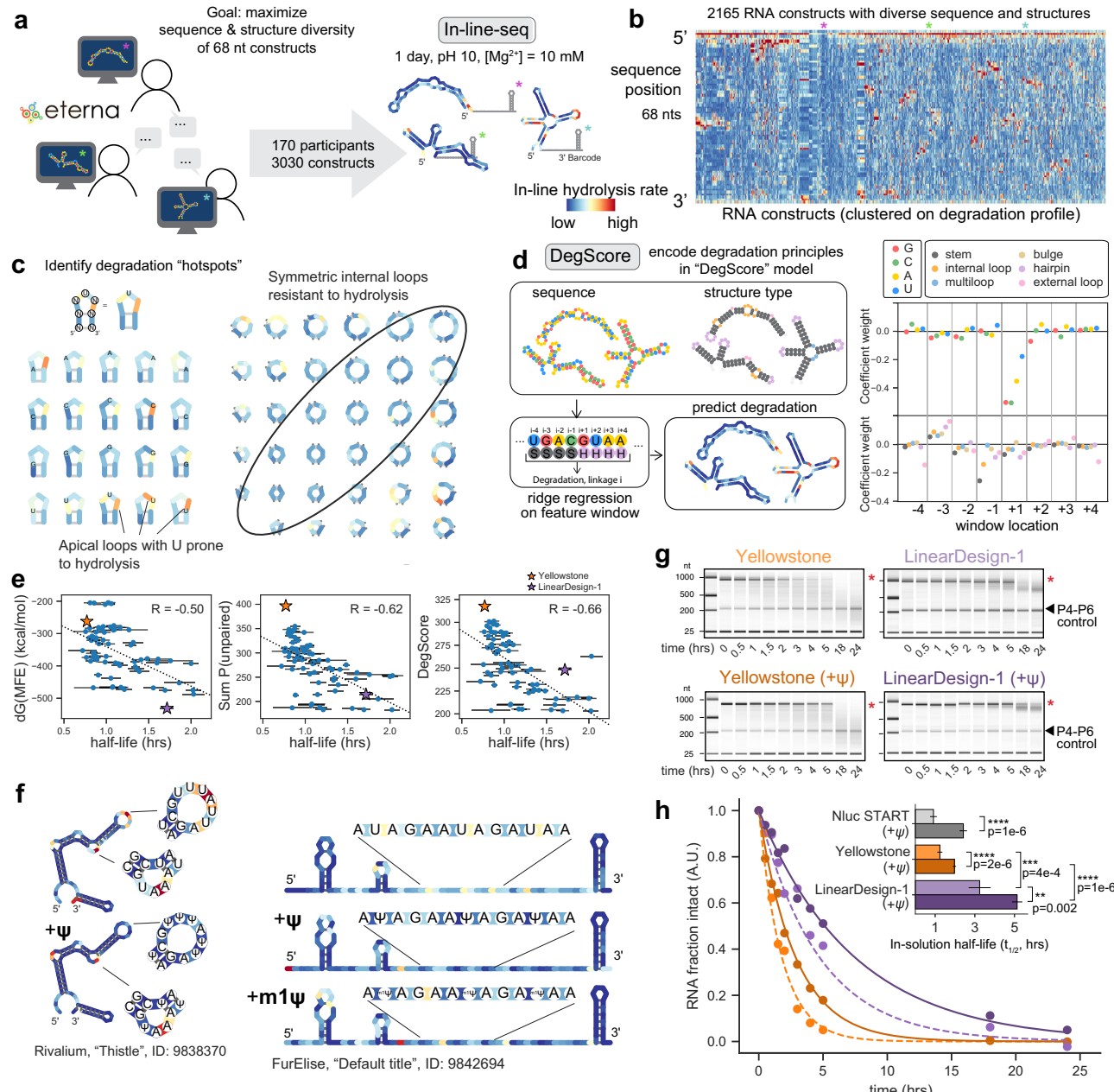

**Fig. 3 High-throughput in-line hydrolysis uncovers principles of in-solution RNA degradation. a** Eterna participants were asked to design 68-nucleotide RNA fragments maximizing sequence and structure diversity. In total, 3030 constructs were characterized and probed using high-throughput in-line degradation (In-line-seq). **b** Nucleotide-resolution degradation of 2165 68-nt RNA sequences (filtered for signal quality), probed by In-line-seq, sorted by hierarchical clustering on degradation profiles. **c** Sequences span a diverse set of secondary structure motifs, revealing patterns in degradation based on both sequence (i.e., linkages ending at 3′ uridine are particularly reactive) and structure (symmetric internal loops, circled, have suppressed hydrolytic degradation compared to asymmetric internal loops). **d** The ridge regression model "DegScore" was trained to predict per-nucleotide degradation from sequence and loop assignment information. Coefficients with the largest magnitude corresponded to sequence identity immediately after the link, with U being most disfavored. **e** DegScore showed improved predictive power on mRNAs over two other metrics previously posited to predict RNA stability. Half-life: in-solution mRNA half-life, calculated from degradation coefficients of the exponential decay fit on time course data in PERSIST-seq. Errors are standard deviations estimated by exponential fits to bootstrapped data. dG(MFE): Free energy of minimum free energy structure, calculated in RNAfold v2.4.14. Sum p(unpaired): Sum of unpaired probability, calculated in RNAfold v2.4.14. **f** Introduction of pseudouridine (ψ) and N1-methylpseudouridine ($^{m1}$ψ) modifications stabilizes selected short RNAs at U nucleotides in both loop motifs and in fully unstructured RNAs. **g** Capillary electrophoresis characterization of fragmentation time courses of Nluc mRNA molecules designed with extensive structure (LinearDesign-1) and relatively less structure (Yellowstone), synthesized with standard nucleotides and with ψ modifications. The full-length mRNA band is indicated with a red asterisk. The *Tetrahymena* ribozyme P4-P6 domain RNA was included after degradation as a control. This result has been repeated independently two times with similar results (cf. Supplementary Fig. 10). **h** Exponential fits of capillary electrophoresis measurements of intact RNA over ten time points confirm significant differences between in-solution lifetimes of LinearDesign-1 and Yellowstone Nluc mRNAs. Inset: Calculated half-lives. mRNA half-life data are presented as mean values ± SD, as estimated from one biological replicate via bootstrapped exponential. Asterisks correspond to two-sided significance tests with ****$p < 0.0001$, ***$p < 0.001$, **$p < 0.01$.

suppression of in-line hydrolysis at the substituted residues, presumably through changed nucleophilicity at the site of substitution (example constructs in Fig. 3f, statistics over all constructs probed by capillary electrophoresis in Supplementary Fig. 5). We also observed suppression of in-line hydrolysis at nucleotides 1 to 2 positions 5′ of the substitution (Fig. 3f), possibly due to locally enhanced base stacking[74]. Structure-mapping data by SHAPE and DMS profiling confirmed that ψ and m[1]ψ substitutions did not change the chemical reactivity of the RNAs outside the substituted positions, consistent with no change in global secondary structure; the suppression of in-line hydrolysis appears to be due to local chemical or structural effects at the site of substitution. However, more detailed measurements are needed to more holistically understand the effect of ψ on secondary structure[8,75].

As a further test of the stabilizing effect of nucleoside modification, we prepared six constructs from the 233x-mRNA library with U or ψ (Fig. 1), including the LinearDesign-1 and Yellowstone RNAs (Fig. 2i). In-solution mRNA half-lives were measured using capillary electrophoresis to evaluate the fraction of intact mRNA over time (Fig. 3g, h, pairwise significance comparisons between constructs in Fig. 3h inset). Consistent with our in-line hydrolysis data on small RYOS RNAs, we observed a 1.2–2.7-fold stabilization for these longer Nluc-encoding mRNAs when U was substituted with ψ (Fig. 3h, Supplementary Fig. 6c). This finding indicated that beyond redesigning RNA sequences to adopt stable structure, in-solution RNA stability can be further improved by incorporating modified U nucleosides.

**Additive effects of UTR, CDS, and ψ improvements**. Thus far we found that highly structured 5′ and 3′ UTRs can support efficient protein synthesis and based on our RNA degradation predictor DegScore and Eterna-derived designs, we saw that the highly structured CDSs can strongly affect in-solution mRNA stability and protein synthesis (Figs. 2, 3). We next investigated if selected UTRs and CDSs in combination with nucleoside modifications may achieve stable and highly translated mRNAs that profit from additive effects of individual improvements. (Fig. 4, Supplementary Fig. 9).

We first determined the combined contribution of different CDS designs and UTRs on mRNA stability and protein expression. It is important to distinguish ribosome load, the weighted sum of mRNA proportions multiplied by the ribosome number in a fraction, from protein output of an mRNA. Protein expression of an mRNA is not directly inferable from its ribosome load, thus luciferase-based expression analysis needs to confirm a positive effect on protein production based on improved mRNA design. We chose six CDS designs and three UTR combinations from our screen (Figs. 1a, 4a, Supplementary Fig. 9). The selected CDS designs were from diverse origins that support a range of in-solution half-lives (from 0.69- to 1.8-fold relative to 'Nluc start'; Fig. 2g). The combined 5′ and 3′ UTRs were individually predicted and/or confirmed (Fig. 2f) to facilitate the highest protein expression in our library (Fig. 2e, f): our standard *hHBB* 5′ and 3′ UTRs; a SARS-CoV-2 5′ UTR dSL-3 variant paired with the dengue virus 3′ UTR, predicted to have high translational efficiency but shorter in-cell half-life due to increased length; and C3 5′ UTR paired with a SINV U-rich element 3′ UTR predicted to have good translational efficiency with a short length (Fig. 2e, f, Supplementary Data 1). In terms of in-solution stability, we expected the mRNAs with the longer UTRs to have reduced half-life across all 6 CDSs, which was confirmed experimentally (Supplementary Fig. 10).

To evaluate protein expression, the 18 mRNAs were individually transfected and luciferase activity was measured after 6 h

(translation rate before significant mRNA decay) and 24 h (total protein output after mRNA decay) (Fig. 4a). After 6 h, two CDS variants (LinearDesign-1 and −423.7) achieved similarly high protein levels compared to Nluc start, when combined with *hHBB* 5′ and 3′ UTRs. The LinearDesign-1 result was striking given its monosome-concentrated polysome profile in PERSIST-seq (Fig. 1e) and the expectation that high mRNA structure should adversely impact the cellular translational apparatus. The result was nevertheless consistent with our model that enhanced in-cell half-life of a structured mRNA may compensate for low translation efficiency (Fig. 2c–e). Indeed, by 24 h, LinearDesign-1 CDS mRNA displayed a two-fold increase in luciferase yield compared to Nluc start (Fig. 4a). LinearDesign-1 CDS mRNAs also exhibited a particularly long in-solution half-life (Supplementary Fig. 10). Among the UTR combinations and CDS designs tested, most demonstrated lower overall luciferase activity than Nluc start with *hHBB* UTRs (Fig. 4a). One exception was the CoV-2-UUG-dSL-3/DEN2 UTR combination, chosen based on its high ribosome load (Fig. 2d, e), which was able to support levels of protein synthesis at 6 h nearly as high as the *hHBB* UTR for the LinearDesign-1 CDS; however, this expression was lowered by 24 h (Fig. 4a). This finding is consistent with faster mRNA decay compared to *hHBB* UTRs and our results supporting that in-cell mRNA stability is a primary driver of protein output (Fig. 2e, e).

Given the improvements in in-cell stability we observed for ψ-modified mRNAs, we further tested the effect of ψ on in-cell stability and protein levels achieved from Nluc start vs. LinearDesign-1 CDS (Fig. 4a). As expected, preparation of mRNAs with ψ led to increased in-solution stability across both these CDSs, independent of UTRs (Supplementary Fig. 10). In terms of protein expression, we observed variable effects on overall luciferase activity by different UTR combinations with fixed CDSs compared to unmodified mRNAs after both 6 and 24 h of expression. Both Nluc start and LinearDesign-1 CDSs with *hHBB* UTRs maintained high protein expression at 6 and 24 h (Fig. 4a) indicating that, despite ψ modification and highly structured CDSs, translation is sustained. Overall, these results demonstrated the importance of mRNA stability to protein output, and suggested that the *hHBB* UTRs, highly structured CDSs, and use of ψ is preferred for increasing in-solution stability and protein output.

**Integration of all design rules leads to mRNAs with both high in-solution stability and high protein output**. For our final experiments, we sought to test whether further optimization, especially of the CDS, might allow both enhanced in-solution mRNA stability and in-cell protein expression. A simple calculation of the number of synonymous mRNAs for the Nluc CDS (621 nt) using a human codon table reveals that there are $1.6 \times 10^{101}$ potential sequence combinations. The hypothesis that mRNA stability may be increased by changing the CDS sequence has been explored theoretically through algorithms designed to optimize the predicted MFE of a CDS[47,76,77], as well a biophysical model for hydrolysis that predicted a minimum two-fold decrease in hydrolysis could be achievable[7], but these predictions for increases in stability have not been experimentally tested. To test these algorithms and aforementioned models in the context of our model mRNAs, we collected a variety of CDS designs to compare to the Nluc start and Yellowstone mRNAs, including (1) the default output of available mRNA design algorithms, including those provided by Genewiz, Twist, and IDT websites and others that may enhance mRNA structure (LinearDesign[47] and use of GC-rich codons[9]), (2) highly structured constructs that were rationally designed through an Eterna competition

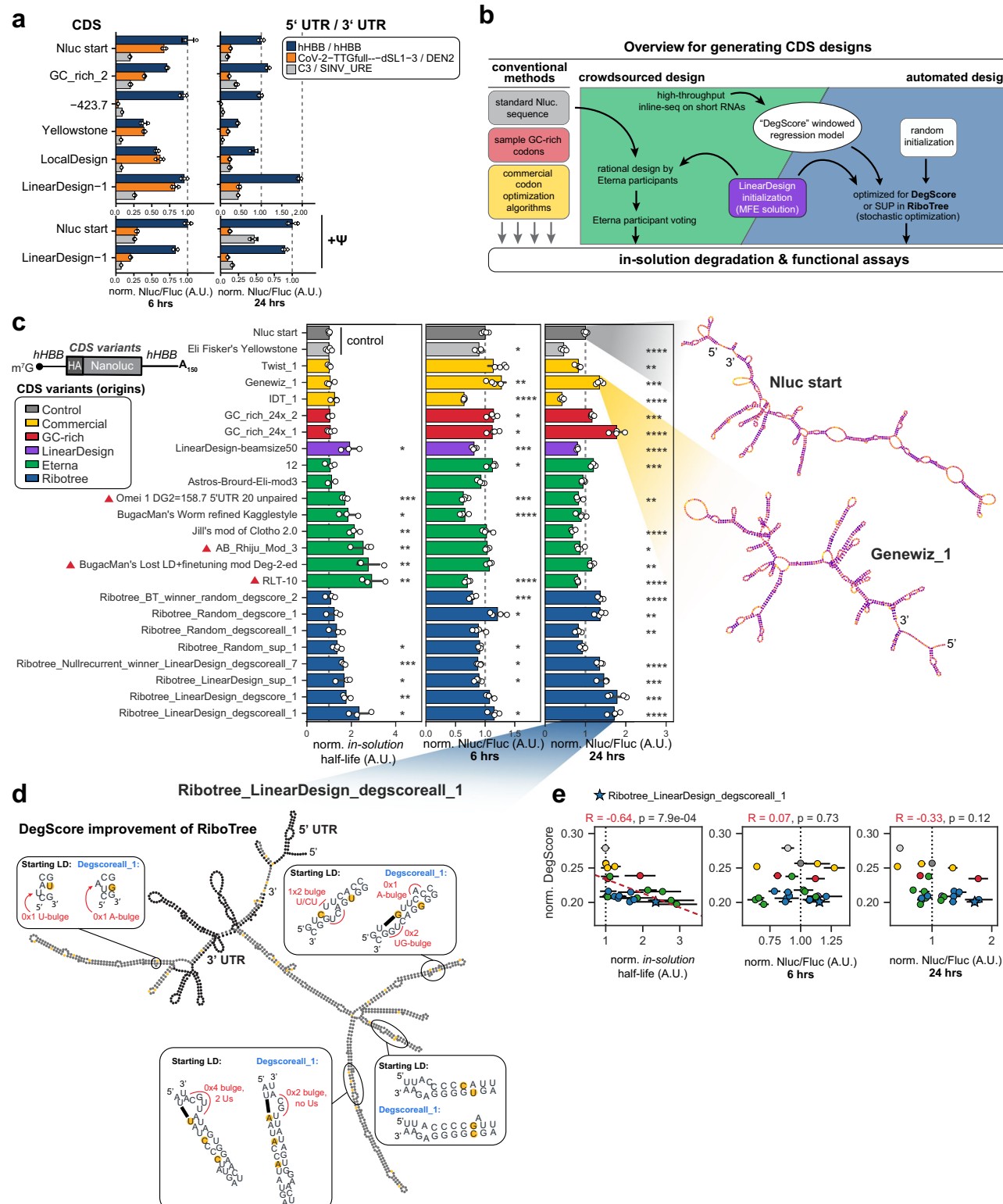

('OpenVaccine: Focus on the NanoLuciferase mRNA'), and (3) output of the automated mRNA structure design tool Ribotree, a stochastic optimization algorithm that can start from different seed sequences (random or LinearDesign) and improve in-solution mRNA half-life as guided by different predictors (AUP, DegScore) (Fig. 4b). From these approaches we generated 24 different CDS designs for Nluc mRNA and, based on our data above, we appended the *hHBB* 5′ and 3′ UTRs as constant regions to mediate high rates of translation initiation and used ψ due to

its enhancement of in-solution mRNA stability (Fig. 3f, g). These 24 different CDS designs displayed a wide variety of predicted structural diversity even for a short ORF such as Nluc (see Supplementary Fig. 11).

Each mRNA design was subjected to accelerated degradation and capillary electrophoresis to measure the in-solution half-life of each individual CDS. Compared to the Nluc start, Eterna-originated designs exhibited up to 2.9-fold higher in-solution half-life (see, e.g., RLT-10) while designs from commercial

**Fig. 4 Integration of 5′/3′ UTRs, structure-optimized CDSs, and pseudouridine (ψ) together enhance mRNA stability and translational output. a** CDS and 5′/3′UTR combinations differentially impact protein synthesis. Six mRNA constructs were in vitro synthesized and luciferase activity was measured 6 or 24 h post-transfection. Inclusion of ψ was tested on two selected constructs. Bars indicate the geometric mean of Nluc/Fluc reporter activity ratios normalized relative to Nluc start/hHBB UTRs. Error bars indicate geometric standard deviation. $n = 4$ biologically independent samples. **b** Workflow for different approaches to design the CDS variants tested in (**c**). **c** Variations in CDS design facilitate high in-solution stability and differential protein expression. In vitro transcribed mRNAs (24 in total) were subjected to in-solution degradation or transfected into HEK293T cells for 6 and 24 h. In-solution half-lives and luciferase activity are normalized to the Nluc start reference construct. Predicted secondary structures are shown for select constructs with colors indicating DegScore at each nucleotide. Designs derived from LinearDesign solutions are marked with a purple triangle. Asterisks correspond to two-sided significance tests with ****$p < 0.0001$, ***$p < 0.001$, **$p < 0.01$. Exact $p$-values are provided in Supplementary Data 5. Bars indicate the mean of Nluc/Fluc reporter activity ratios normalized relative to Nluc start. Error bars indicate standard deviation across $n \geq 3$ biologically independent samples. **d** Predicted secondary structure overview of Ribotree_LinearDesign_degscoreall_1. Zoomed boxes indicate sequence optimizations and subsequent structural changes made by DegScore to the reference LinearDesign construct. **e** Increased in-solution half-life correlates with DegScore. Significance test for Spearman correlation value: two-sided $p$-value for a hypothesis test whose null hypothesis is that two sets of data are uncorrelated, $n = 24$. Error bars indicate standard deviation across $n \geq 3$ biologically independent samples.

algorithms or optimized GC-content exhibited similar in-solution half-lives (Fig. 4c, significance values in Supplementary Data 5). Designs from the LinearDesign server and modifications to these designs from both Eterna participants (e.g., AB_rhiju_mod3) and the RiboTree algorithm produced constructs with increased half-life up to 2.3-fold over Nluc (triangles in Fig. 4c). When individual mRNA designs were assayed for protein expression, despite overall longer in-solution half-lives, at 6 h, both Eterna- and Ribotree-derived designs had similar or lower luciferase activities than Nluc start, whereas most vendor-derived and GC-rich designs had slightly higher activities (Fig. 4c). At 24 h, the trend of lower luciferase activity remained for 6 of 8 Eterna-derived designs tested (Fig. 4c, green). However, in contrast, 6 of 8 RiboTree-optimized mRNA designs demonstrated higher luciferase activities than Nluc start at 24 h (Fig. 4c, blue). The sequence output by RiboTree starting from a LinearDesign CDS solution, with optimization guided by DegScore and flanking *hHBB* UTR sequences, yielded an mRNA that was both highly stable in solution ($t_{1/2} = 2.3$, relative to Nluc start) and exhibited high levels of protein expression (1.7-fold increase, relative to Nluc start) (Ribotree_LinearDesign_degscoreall_1; Fig. 4c). This simultaneous increase in in-solution stability with improved and sustained in-cell protein expression provided a strong demonstration of the impact of rational design for mRNA.

To gain insight into what led to success for this RiboTree_LinearDesign_degscoreall_1 sequence, we examined the specific regions RiboTree modified from the starting sequence as it computationally minimized the DegScore metric from a starting LinearDesign server solution (Fig. 4d, with further comparison in Supplementary Fig. 11). These computational modifications are characterized by reducing the presence of Us in loops, and shifting local base-pairing to minimize the overall size of loops, even if such shifts result in additional smaller loops. These modifications are consistent with mitigating hydrolysis as modeled in the DegScore predictor (Fig. 3d). Taken together, these data suggest that by reducing the overall presence of Us in loops and reducing the number of hairpins to generate a "linear" highly double-stranded mRNA can lead to enhanced mRNA stability and protein expression.

Correlating data from these final 12 Nluc RNAs provided additional insight into the biophysical features impacting mRNA performance (Fig. 4, Supplementary Fig. 12). Most notably, DegScore (adjusted to account for reduced PSU reactivity, see Methods) correlated strongly with measured in-solution half-life ($R = -0.64$, $p < 0.001$), and moderately with 24-h protein expression ($R = -0.33$, $p = 0.12$), yet was uncorrelated with 6-h protein expression (Fig. 4e). However, 6-h protein expression correlated strongly with the predicted number of hairpins ($R = 0.59$, $p < 0.001$) and the "Maximum Ladder Distance[78]," or

the maximum helix path length ($R = -0.80$, $p < 0.001$) (Supplementary Fig. 12c, Supplementary Data 6). These observations suggest that resistance to RNA hydrolysis, as predicted by DegScore and quantified by in-solution half-life, is important for longer protein expression, but that other RNA sequence and structural features govern protein expression at shorter time-scales. For instance, longer or more branched double-stranded RNA stems may potentially hinder the ability of the ribosome to unwind RNA secondary structure.

In all our tests above, we measured in-solution half-lives of mRNAs using structural readouts (RT-PCR; capillary electrophoresis), but sustaining functional output after degradation is of the strongest interest for mRNA applications. As a final experiment, we, therefore, carried out an experimental stress test of in-solution stability with a functional readout based on transfection and protein production in cells. For this 'end-to-end' test of mRNA efficacy, we synthesized Nluc start mRNA alongside the optimized Ribotree_LinearDesign_degscoreall_1 mRNA both with U or ψ. Each individual Nluc mRNA was then subjected to in-solution degradation and 8 time points were collected (Fig. 5a). The optimized Ribotree_LinearDesign_degscoreall_1 mRNA exhibited higher resistance to degradation in-solution than the Nluc start mRNA and incorporation of ψ into either mRNA further enhanced in-solution stability (Fig. 5b). The majority of the mRNA stabilized with ψ and a structure-optimized CDS remains functional after 2 h of accelerated solution degradation, whereas the starting mRNA sequence gives negligible in-cell activity after the same time of degradation (Fig. 5b).

To test the reproducibility of the results above, mRNAs (Fig. 4c) generated by the Das and Barna groups at Stanford were shared with Pfizer's viral vaccine group to compare the in-solution stability and protein expression in tissue culture (Fig. 5c–e). Compared to the Nluc start, four optimized CDS design RNAs were tested for in-solution stability when complexed with a cationic polyplex molecule (PLX) (Fig. 5c, Supplementary Fig. 13). After an extended in-solution incubation period at 5 °C, both the Eterna-derived (RLT-10 and BugacMan's_Lost_LD + finetuning_mod_Deg-2-ed) and Ribotree-derived (Ribotree_LinearDesign_degscoreall_1) designs had longer half-lives than the GC-rich (Genewiz) and reference (Nluc start) sequences (Fig. 5d). The level of expression from the mRNAs in HEK293T cells at the end of a 2-weeks incubation had decreased less for the Eterna-derived (18% decrease from time 0 for RLT-10 and 8% for BugacMan's_Lost_LD + finetuning_mod_Deg-2-ed) and Ribotree-derived (2%) designs than for the GC-rich (32%) and reference Nluc start RNAs (43%) (Fig. 5e).

Taken together, our data indicate that combining translation-facilitating UTRs with structural optimization of CDSs via

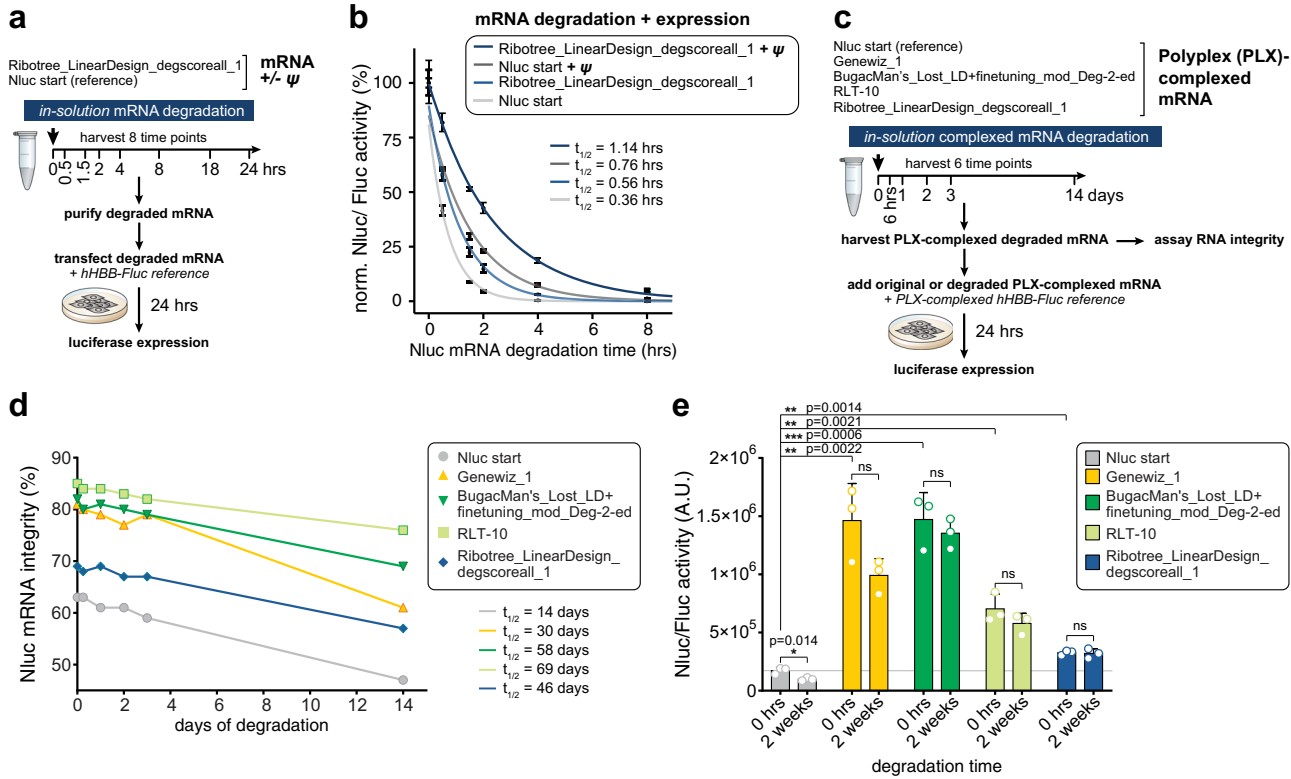

**Fig. 5 Stability and cellular expression of selected highly structured RNA designs in solution and formulated with polyplex. a** Schematic for testing the synergy between RNA modifications and mRNA design rules on downstream stability and protein output. mRNAs were in vitro synthesized with or without ψ and subjected to degradation conditions. Samples were collected overtime and the RNA was purified before being transfected into HEK293T cells. Luciferase activity was measured 24 h after transfection. **b** Luciferase activity of the reference Nluc sequence and DegScore-optimized CDS with or without ψ after being subjected to in-solution degradation. mRNA half-lives ($t_{1/2}$) per construct are given in hours (hrs). Plotted on y-axis are the geometric mean of Nluc/Fluc reporter activity ratios normalized to time zero. Error bars indicate geometric standard deviation. $n = 4$ biologically independent samples. **c** Schematic for testing the effect of RNA formulation on downstream stability and protein output from selected RNA designs. mRNAs were in vitro synthesized, formulated with polyplex (PLX), and subjected to degradation conditions and/or expression analysis. Samples were collected over time and the formulated RNA was added to HEK293T cells. **d** In vitro stability of RNAs formulated with polyplex over 14 days at 5 °C. RNA half-lives were calculated based on the degradation slopes: Nluc start (reference) (14 days), Genewiz_1 (30 days), BugacMan's_Lost_LD + finetuning_mod_Deg-2-ed (58 days), RLT-10 (69 days) and Ribotree_LinearDesign_degscoreall_1 (46 days). Results correspond to technical duplicates. **e** Expression of Nluc from HEK293T cells transfected with selected RNA designs formulated with polyplex. Expression was measured by fluorescence after the RNAs were formulated with polyplex, incubated at 5 °C in degradation conditions for 0 and 14 days, and then added to the medium of the cultured cells. Results correspond to technical replicates; normalized Nluc/Fluc activity ± SD. $n = 3$; ns not significant. *$p \leq 0.05$ was considered significant (two-tailed unpaired Student's t-test; ns: $p > 0.05$; *$p \leq 0.05$; **$p \leq 0.01$; ***$p \leq 0.001$; ****$p \leq 0.0001$).

LinearDesign and DegScore-guided RiboTree design, we can enhance both in-solution stability and total protein output of mRNAs. Moreover, downstream luciferase expression can be further amplified by ψ modification of mRNAs.

## Discussion

mRNA-based therapeutics are transformative in the way in which human disease is treated, such as infectious disease (e.g., the timely COVID-19 mRNA vaccines)[79–82]. However, there still remain several major hurdles for mRNA-based therapeutics to be effective, many of which are directly linked to the intrinsic features of mRNA molecules. RNA is inherently unstable due to its 2′-hydroxyl group, and degradation of the mRNA in solution by in-line nucleophilic attack, potentially exacerbated by lipid formulations, poses a major challenge[6]. Moreover, once delivered into patient cells, the candidate mRNA must outcompete other cytoplasmic mRNAs for the translation machinery and avoid the cellular mRNA degradation machinery to express maximum amounts of the desired encoded protein. Our study presents advances in tackling this challenge, building on two RNA-seq-based technologies.

Our first set of experimental findings derive from the integrated PERSIST-seq technology, which enables parallel evaluation of the effects of UTR and CDS sequence and structure on in-cell mRNA translation efficiency, in-cell mRNA stability, and in-solution stability. This technology identifies previously unknown tradeoffs and additive effects between different tunable aspects of mRNA performance as key determinants. Most important for enhancing these characteristics of mRNA performance and their interdependencies, we find that in-cell mRNA stability may be a greater driver of protein output than high ribosome load. This might particularly be important when proteins need to be expressed from a single dose of mRNAs for long periods of time. Specifically, mRNAs associated with the heaviest polysomes, thought to be linked to the highest translation efficiency, were found to be less stable: there may be a 'sweet spot' of ribosome loading and translation efficiency to achieve maximal total protein output. This effect may derive from overcrowding of ribosomes on mRNA coding regions due to efficient and rapid initiation, which can induce ribosome queuing and sterical ribosome collisions that have recently been found to lead to translation-dependent mRNA decay[83,84]. In turn, mRNAs may be

more stable with an optimal number of ribosomes loaded for efficient protein synthesis. A potential explanation for a highly structured CDS still being well translated could be the strong intrinsic helicase activity of the ribosome while translocating along mRNAs[85], which can unwind stable CDS structure in a two-step mechanism inside the elongating ribosome[86], without detriment to protein output. We also cannot rule out the potential contribution of cellular RNA helicases that may play a role in restructuring or unwinding structured CDSs[87]. Overall, we observed a wide range of UTR-dependent translation efficiencies in PERSIST-seq which could be further fine-tuned with specific UTR sequence or CDS structure alterations. We note that for applications with very short mRNAs (as used for MEVs, and represented in our MEV sequences[88]) or much longer mRNAs (as are needed for antigens like the SARS-CoV-2 Spike protein), the best UTRs and the sweet spot for optimal translation efficiency may be different. In this respect, certain UTR combinations outperform *hHBB* in translation efficiency and include multiple cellular 5′ UTRs as well as, unexpectedly, the dengue virus 5′ and 3′ UTRs, which both individually increase ribosome loading, and combining them in one mRNA resulted in an additive effect. As illustrated with the SARS-CoV-2 5′ UTR, selective translational enhancers can be further identified through careful mutagenesis and deletion strategies aimed at narrowing selective regulatory regions within viral leader sequences. We also note that our experiments were mostly limited to HEK293T as model human cells; the best UTRs for high protein output are likely to be cell-type dependent and therefore will depend on the application. Our library of UTRs and the PERSIST-seq technology should be well-suited for discovering and leveraging enhanced UTRs for future applications with different protein targets and cell types.

Our second set of experimental improvements relate to mRNA stability in aqueous solution. PERSIST-seq confirms that extremely highly structured mRNAs can exhibit more than double the in-solution half-lives of conventionally designed mRNAs, with strong implications for improved storage in solution. Further design insights came from a newly developed method called In-line-seq for high-throughput in-line probing[11,66], applied to thousands of diverse Eterna-derived short RNAs. The In-line-seq results reveal a number of structural and simple sequence rules for mitigating in-solution RNA hydrolysis: a key determinant of in-solution RNA degradation is the presence of uridine and that RNA linkages 5′ of a uridine residue are particularly susceptible to degradation, which can be alleviated through the inclusion of ψ or m¹ψ. Synthesis of mRNAs with modifications—already in wide use for crucially mitigating innate immune response and translational shutdown by mRNA therapeutics[69,70,79,80]—is therefore a simple additional improvement to achieve greater in-solution stability while sustaining protein expression.

Further leveraging our in-line hydrolysis data, we developed DegScore, a model for hydrolytic degradation that was independently validated on PERSIST-seq data and which enables in silico optimization of any RNA sequence. By combining optimal UTRs, DegScore optimization with RiboTree, and ψ modification, we are able to achieve high mRNA stability and improved protein expression. We note that DegScore was trained on degradation from unmodified nucleotides that does not account for our observed stabilization via ψ. Thus, future studies training similar models on degradation data from ψ and other nucleoside modifications may result in improvements to CDS stabilization via algorithmic design.

Ultimately, by applying DegScore, pseudouridine, and our insights from PERSIST- and Inline-seq we designed an mRNA that demonstrated a ~1.7-fold increase in protein output and ~2.5-fold increase of in-solution half-life. While it is difficult to directly assess the exact contribution of each mRNA design

element given the confounding variable that each element presents, our results suggest that a given 5′ or 3′ UTR has a greater influence on ribosome load, while the inclusion of a structured CDS greatly influences the stability of the mRNA both in-cell and in-solution. Thus, it is likely the combination of both of these that allowed for the design of a highly translated and stable mRNA.

We also note that our study has focused on characterizing degradation of non-formulated mRNAs in order to understand limitations on stability imposed by the fundamental biophysical and biochemical properties of RNA; such non-formulated mRNAs also appear optimal for certain applications including personalized cancer mRNA vaccines[89]. For applications that benefit from mRNA formulated in lipid nanoparticles or other carriers, we expect future studies applying PERSIST-seq and In-line-seq to formulated mRNA libraries to reveal additional insights. Lastly, it has been proposed that highly structured mRNAs may retain their structure and in-solution stability under temperature shifts, mutations, and changes in UTRs, motivating the term "superfolder" mRNAs[7]; it will be interesting to test these predictions through future PERSIST-seq studies. These results demonstrate a modular and flexible platform that is applicable to potentially any protein target of choice and can accelerate the design of overall improved mRNA therapeutics solutions.

Overall, we report an mRNA design methodology that can enhance mRNA stability in solution while sustaining or even increasing protein expression inside cells. There does not have to be a tradeoff between mRNA structure, stability, and protein output. Looking ahead, our computational-experimental platform can be rapidly developed to customize highly structured mRNAs for target proteins. As mRNA-based medicines are explored for a wide range of human diseases including cancer therapies, we hope that these insights and methods can help these medicines become more effective, manufactured at a lower cost per patient, and more accessible and widely distributed to alleviate disease.

## Methods

**In vitro transcription of reporter mRNAs.** The preparation of mRNAs were based on in vitro transcription from DNA templates. DNA templates were amplified by PCR using AccuPrime Pfx (Life Technologies, 12344024) and purified using the Monarch PCR & DNA Cleanup Kit (NEB, T1030L). The source of the 3xHA-Nluc starting CDS ("Nluc start") is derived from the pcDNA3.1-5′UTR-3xHA-Nluc plasmid encoding the HA-tagged Nanoluc CDS[44]. Individual template DNA or the 233-mRNA library was amplified from linear DNA synthesized on a BioXP 3200 system (Codex DNA) or by Twist Bioscience, using the fixed forward (T7_F_28nt) and reverse (const3_R) primer. The forward primer binds to the T7 RNA polymerase promoter common in DNA template for all mRNA designs; the reverse primer is complementary to a common "const3" region at the end of all tested mRNA 3′ UTRs. For the IVT template pool, individual DNA templates were pooled for a template pool of hundreds of constructs at an equimolar concentration and are amplified with outer primers in a pooled format. For the pooled template, 1 μL of each construct (~20 ng/μL stock concentration) was pooled to be used as the PCR template. The Pfx PCR contained the following: 2.5 μL 10x Pfx buffer, 0.25 μL forward primer (100 μM), 0.25 μL reverse primer (100 μM), 0.75 μL DMSO (NEB), 0.25 μL Pfx Polymerase (Thermo), 20.5 water, and 0.5 μL template DNA (~20–50 ng/μl), in a total 25 μL reaction with the following program: 2 min at 95 °C; 10 s at 95 °C; 30 s at 58 °C; 30 s or 1 min at 68 °C; cycled 9x; final extension of 5 min at 68 °C. PCR reactions were purified with Monarch PCR & DNA Cleanup Kit (NEB, T1030L). For the hHBB-Fluc control mRNA, the DNA template was amplified from the pGL3-HBB plasmid[90] using the primers KL588/KL589 which yielded a PCR product of 1750 kb in length. For cloning the MALAT1 ENE 3′ UTR stem-loop, we first amplified the ENE region using primers ENE-1/ENE-2 with flanking constant regions. The resulting amplicon was assembled with a hHBB-Nluc sequence that lacked a 3′ UTR but maintained a unique barcode using a NEBuilder HiFi Assembly Kit (NEB, ES2621).

In vitro transcription was performed with the MEGAscript T7 kit (Ambion, AM1333) according to the manufacturer's instructions. A 20 μL transcription reaction contained max. 5 μg linear DNA template, 4 mM of each NTP (Ambion), 2 μL/200 U MEGAscript T7 RNA polymerase (Ambion) and 1x T7 MEGAscript Transcription Buffer (Ambion). After a total incubation for 3 h at 37 °C, the DNA was digested by addition of 1 μL/2 U Turbo DNase (Ambion, AM2238) for 15 min at 37 °C. For pseudouridylated mRNAs, pseudouridine triphosphate (Trilink Biotechnologies, N1019-5) was substituted for uridine triphosphate at an

equivalent concentration. mRNA was purified using MegaClear columns (Thermo Scientific, Ambion, AM1908). A 20 µL reaction usually yielded 100–150 µg of RNA.

For mRNA transfection of HEK293T cells, m7G-capped and polyadenylated mRNAs were generated as follows. In vitro transcribed mRNA was then m7G-capped and polyadenylated using the ScriptCap m7G Capping System (CellScript, C-SCCE0625) and A-Plus Poly(A) Polymerase Tailing Kit (CellScript, C-PAP5104H), respectively, according to the manufacturer's instruction with the following modifications. Aliquots of 30 µg of each RNA were processed in parallel, diluted to 34.25 µL in water and heated for 5 min at 65 °C to denature and placed on ice. The 50 µL capping reaction contained 5 µL 10x ScriptCap buffer (Cellscript), 5 µL 10 mM GTP (Cellscript), 2.5 µL 2 mM S-adenosyl-methionine (SAM, 20 mM stock, Cellscript), 1.25 µL ScriptGuard RNase Inhibitor (Cellscript), and 2 µL Capping enzyme (20 U, Cellscript, 10 U/µL). For the capping step, the 37 °C incubation was performed for 1 h and the capped RNA was placed on ice. Polyadenylation was performed from the resulting RNAs without purification in between. The polyA reaction contained 30 µg of capped RNA in 50 µL, 6.6 µL 10x A-Plus polyA tailing buffer (Cellscript), 6.6 µL 10 mM ATP (Cellscript), 0.3 µL ScriptGuard RNase Inhibitor (Cellscript), and 2.5 µL A-Plus PolyA Polymerase (10 U, 4 U/µL, Cellscript) in a total reaction volume of 66 µL. We aimed to add a 150 nt-long polyA-tail for which we incubated the capped mRNA for 30 min at 37 °C with 10 U of polyA enzyme, after which the reaction was placed on ice. The mRNA was again purified using MegaClear columns. mRNA concentration was determined on a Nanodrop 2000 (Thermo Fisher). This usually yields 30–40 µg of capped and polyadenylated mRNA. mRNA quality was determined by 4% urea-PAGE, 1% formaldehyde agarose gel or capillary electrophoresis with an Agilent 2100 Bioanalyzer (Agilent Technologies). A list of all primer sequences used are provided in Supplementary Data 7. Raw data images of gels presented are given in the Source Data.

**Formulation of RNAs in Polyplex (PLX).** In vivo-jetPEI® was purchased from Polyplus Transfection®. All other chemicals were multi-compendial grade and procured from Sigma-Aldrich and Fisher. Polyplex was prepared by mixing RNA and in vivo-jetPEI® in optimized buffer matrix. Briefly, in a 2 cc sterile glass vial, RNA stock solution was mixed with HEPES buffer and Glucose. PEI solution was prepared with histidine buffer and glucose in another sterile glass vial. Two parts were mixed in 1:1 ratio by volume, followed by 10-time inversion by hand. The final matrix was 5 mM histidine, 5 mM HEPES, and 5% w/v glucose, with N:P ratio of 12 and final pH 6.0. Samples were then aliquoted and transferred to 5 °C for stability study.

**Cell culture and transfections.** HEK293T (ATCC: CRL-3216) cells were cultured in Dulbecco's Modified Eagle's Medium (DMEM, Gibco, 11965–118) containing 2 mM L-glutamine, supplemented with 10% fetal bovine serum (EMD Millipore, TMS-013-B), 100 U/ml penicillin, and 0.1 mg/ml streptomycin (EmbryoMax ES Cell Qualified Penicillin-Streptomycin Solution 100X; EMD Millipore, TMS-AB2-C or Gibco, 15140–122) at 37 °C in 5% CO2-buffered incubators. For transfection of pooled 5′ m7G-capped and poly(A)-tailed RNAs, $5.0 \times 10^6$ HEK293T cells were seeded in a 10 cm plate 24 h before transfection. 10 µg of pooled RNAs were transfected with Lipofectamine MessengerMax as per the manufacturer's instructions (Life Technologies). Media was changed 3 h after transfection and replaced with complete DMEM supplemented with 10% FBS and Pen/Strep. For transfections of individual m7G-capped RNAs, $3.0 \times 10^4$ HEK293T cells were seeded per well 24 h before transfection in a 96-well plate. Subsequently, 10 ng of Nluc RNA was co-transfected with 20 ng of m7G-capped HBB-Fluc control RNA using Lipofectamine MessengerMax as per the manufacturer's instructions (Life Technologies). A list of all primer sequences used are provided in Supplementary Data 7. All oligonucleotides were purchased from IDT.

To measure expression from RNA complexed with polyplex, HEK293T cells were seeded at $4.0 \times 10^4$ cells/well in flat-bottom 96-well plates (Corning, Cat# 3596) in 100 µl volume and incubated at 37 °C, 5% CO2 overnight. Samples with an estimated 10 ng of Nluc RNA that had been formulated with polyplex and incubated under degradation conditions for 0 or 14 days were spiked with equal amounts of Fluc RNA in an equal volume. The mixture was directly added to the cell culture media and incubated for a total of 24 h without media change. Transfections were done in triplicates.

**Sucrose gradient fractionation analysis.** Cell culture media was replaced with cycloheximide (MilliporeSigma, C7698-1G) containing media at 100 µg/mL. After 2 min, cells were washed, trypsinized and harvested using PBS, trypsin, and culture media containing 100 g/mL cycloheximide. ~$10 \times 10^6$ cells were resuspended in 400 µL of following lysis buffer on ice for 30 min, vortexing every 10 min: 25 mM Tris-HCl pH 7.5, 150 mM NaCl, 15 mM MgCl2, 1 mM DTT, 8% glycerol, 1% Triton X-100, 100 µg/mL cycloheximide, 0.2 U/µL Superase-In RNase inhibitor (ThermoFisher Scientific, AM2694), 1x Halt protease inhibitor cocktail (Thermo-Fisher Scientific, 78430), 0.02 U/µL TURBO DNase (ThermoFisher Scientific, AM2238). After lysis, nuclei were removed by two step centrifuging, first at $1300 \times g$ for 5 min and second at $10,000 \times g$ for 5 min, taking the supernatants from each. 25–50% sucrose gradient was prepared in 13.2 mL ultracentrifuge tubes (Beckman Coulter, 331372) using Biocomp Gradient Master with the following

recipe: 25 or 50% sucrose (w/v), 25 mM Tris-HCl pH 7.5, 150 mM NaCl, 15 mM MgCl2, 1 mM DTT, 100 µg/mL cycloheximide. The lysate was layered onto the sucrose gradient and ultracentrifuged on the Beckman Coulter SW-41Ti rotor at 40,000 rpm for 150 min at 4 °C. The gradient was density fractionated using Brandel BR-188 into $16 \times 750$ µL fractions, and in vitro transcribed spike-in RNA mix (120002B1, 120010B1, 220023B1, 310333T3; 1000, 100, 10, 1-fold dilutions respectively) were added to each fraction. Seven hundred microliters of each fraction was mixed with 100 µL 10% SDS, 200 µL 1.5 M sodium acetate, and 900 µL acid phenol-chloroform, pH 4.5 (ThermoFisher Scientific, AM9720), heated at 65 °C for 5 min, and centrifuged at $20,000 \times g$ for 15 min at 4 °C for phase separation. Six hundred microliters aqueous phase was mixed with 600 µL 100% ethanol and RNA was purified on silica columns (Zymo, R1013).

**Luciferase activity assay after mRNA transfection.** Media from transiently transfected HEK293T cells was aspirated and cells were lysed in 40 µL of 1x passive lysis buffer from the Dual-Luciferase Reporter Assay System (Promega, E1980) and either directly assayed or frozen at −20 °C. After thawing, 20 µL of supernatant was transferred to a new plate and assayed for luciferase activity using the Nano-Glo Dual-Luciferase Reporter Assay System (Promega, N1610) to measure Firefly (Fluc) and NanoLuc (Nluc) luciferase activities. In particular, 50 µL of ONE-Glo Ex Reagent was added to each well of lysate and incubated for 3 min at room temperature before measuring Fluc activities. Subsequently, 50 µL of NanoDLR Stop & Glo reagent was added to each well, and incubated for 10 min at room temperature before measuring luciferase activities on a GloMax-Multi (Promega) plate reader. Luciferase reporter activity is expressed as a ratio between Nluc and Fluc. Each experiment has performed a minimum of three independent times. Because this assay relies on the accumulation of luciferase in the cytosol, any signal peptide sequences (Supplementary Data 1) were removed from the CDS for templates and mRNA for these transfection and luciferase activity experiments.

At Pfizer, luciferase activity was measured similarly. At the indicated time points, Nluc and Fluc activities were measured using Promega's Dual-Luciferase Reporter Assay System (Cat# E1910). Briefly, supernatant was aspirated and 50 µL of Passive Lysis Buffer was added to each well and incubated at room temperature for 15 min. Fluc activity was measured from 20 µL of lysate in 100 µL of Luciferase Assay Reagent II. Then, 100 µL of Stop & Glo Reagent was added and Nluc activity was measured. Nluc activity was normalized to the Fluc signal in each well and data were reported as Nluc/Fluc normalized to the Nluc start reference RNA.

**Quantitative RT-PCR (RT-qPCR) analysis.** RNA-transfected HEK293T cells were first lysed and separated by sucrose density gradient fractionation as described above. From each fraction. RNA was purified by acidic phenol/chloroform followed by isopropanol precipitation. 0.5 mg of RNA was converted to cDNA using iScript Supermix (Bio-Rad, 1708840). cDNA was synthesized from 100-200 ng of total RNA using iScript Supermix (Bio-Rad, 1708840) containing random hexamer primers, according to the manufacturer's instructions. PCR reactions were assembled in 384-well plates using 2.5 µL of a 1:4–1:5 dilution of a cDNA reaction, 300 nM of target-specific primer mix, and the SsoAdvanced SYBR Green supermix (Bio-Rad, 1725270) in a final volume of 10 µL per well. Data were analyzed and converted to relative RNA quantity using CFX manager (BioRad). For sucrose gradient fractions, the amount of RNA from individual fractions was expressed as a fraction of the total RNA collected from all fractions. Primers were used at 250 nM per reaction. A list of all primer sequences used for qPCR are provided in Supplementary Data 7.

**In cell and in-solution RNA degradation time courses.** For in-cell RNA stability, the 233-member in vitro transcribed mRNA pool (m7G-capped and polyA) was transfected into HEK293T cells as described above and RNA was harvested at 1, 7, 12, and 24 h in Trizol (ThermoFisher Scientific, 15596026). RNA was extracted from the aqueous phase on silica columns (Zymo, R1013).

For in-solution RNA degradation experiments, 750 ng of the 233-mRNA pool (not m7G-capped or polyA) was incubated in 30 µL of Degradation Buffer (50 mM CHES at pH 10 and 10 mM MgCl2) and collected over 10 time points: 0, 0.5, 1, 2, 3, 4, 5, 6, 16 and 24 h. To each sample, 15 µL of 0.5 M Tris-HCl pH 7 and 3 µL of 0.5 M EDTA-Na was added to quench the degradation. The integrity of each sample was checked by loading 5 µL of total RNA alongside a spike-in control (P4P62HP, 50 ng) onto a PAGE-Urea-TBE gel and visualized by SYBR Gold (Thermo Fisher). Subsequently, RNA was purified using Ampure beads + 40% polyethylene glycol 8000 (7:3) and checked again by PAGE-Urea-TBE gel and visualized by SYBR Gold.

**Measurement of RNA Stability by capillary electrophoresis for RNAs formulated in Polyplex (PLX).** All chemicals and reagents other than Fragment Analyzer kits (Agilent) were purchased from Sigma Aldrich and Fisher Scientific. The PLX-RNA samples were aliquoted and incubated at 5 °C. At each time point (0 h, 6 h, 1 day, 2 days, 3 days, and 14 days), sample aliquots were taken and stored at −70 °C. Aliquots were thawed at the same time, and RNA integrities were determined using an Agilent 5300 Fragment Analyzer system (Agilent, CA) in duplicate. Briefly, 45 µL of PLX-RNA samples were mixed with 5 µL of heparin solution (20 g/L heparin, 10 mM HEPES, 1 mM EDTA, pH 7.4), and the mixture

was incubated at 30 °C for 20 min. These processed samples were further diluted 200-fold using Fragment Analyzer high sensitivity RNA diluent maker (DNF-300-0004, Agilent, CA). The diluted samples were loaded onto the Fragment Analyzer (FA). The FA analysis was performed using an RNA high sensitivity kit (DNF-472-0500, Agilent, CA), following the vendor-recommended testing protocol. The key experimental parameters are: Injection voltage — 7 kV; injection time — 100 s; separation voltage — 6 kV; separate time — 90 min. The FA electropherograms were processed with Prosize software (v3.0.1.6, Agilent, CA). The RNA degradation rate was modeled with first-order reaction kinetics (RNA integrity vs time) as shown in the following equation:

$$[Integrity] = [Integrity]_0 e^{-kt} \quad (1)$$

where [Integrity] is %integrity of the RNA tested by Fragment Analyzer at different time points;

[Integrity]$_0$ is % integrity of RNA tested by Fragment Analyzer for $t = 0$ samples; $k$ is the reaction constant of a first-order reaction; $t$ is reaction time. The half-life of RNA in PLX platform was calculated as:

$$t_{1/2} = \frac{ln(2)}{k} \quad (2)$$

where $t_{1/2}$ is the half-time of RNA integrity (50% degradation of RNA as from initial time point).

**Library preparation and amplicon sequencing**. Up to 250 ng RNA in 2.75 μL was mixed with 0.25 μL 2 μM RT_Const2_N12_Read1Partial (Supplementary Data 7) and 0.25 μL 10 mM dNTPs each. The RNA samples were then denatured at 65 °C for 5 min and chilled to 4 °C. 1.75 μL reverse transcription mix was added to 5 μL total reaction volume: 1 μL 5x Superscript IV buffer, 0.25 μL 10 mM DTT, 0.25 μL Superase-In (ThermoFisher Scientific, AM2694), 0.25 μL Superscript IV (Thermo 18091050). The reaction was incubated at 55 °C for 45 min and inactivated at 80 °C for 10 min.

First round PCR was performed under following conditions: 1 μL RT reaction, 10 μL 2x Q5 Hot Start Master Mix (NEB M0494S), 0.2 μL 100x SYBR (Thermo S7563), 1 μL 10 μM Read1Partial_F, 1 μL 10 μM 50:50 Hbb_Fwd:Nluc_Fwd mix in 20 μL total volume. Cycling conditions were: 98 °C for 60 s, and 15 cycles of 98 °C for 10 s, 68 °C for 10 s and 72 °C. Second round PCR was performed under the following conditions: 1 μL first round PCR, 10 μL 2x Q5 Hot Start Master Mix, 0.2 μL 100x SYBR, 1 μL 10 μM Read1Partial_F, 1 μL 10 uM Read2Partial_Const1_R in 20 μL total volume. Cycling conditions were: 98 °C for 60 s, and 5 cycles of 98 °C for 10 s, 72 °C for 5 s. Sequencing adaptors were added using the following conditions for final round PCR: 1 μL second round PCR, 10 μL 2x Q5 Hot Start Master Mix, 0.2 μL 100x SYBR, 1 μL 10 μM NEBNext Index Primer (NEB E7335, NEB E7500, NEB E7710, NEB E7730, NEB E6609), 1 μL 10 μM NEBNext Universal PCR Primer in 20 μL total volume. Cycling conditions were: 98 °C for 60 s, and 5 cycles of 98 °C for 10 s, 72 °C for 5 s. All barcoded samples were then pooled at equal volumes and purified with 1.1x SPRIselect beads (Beckman Coulter B23317). Sequencing was performed at the Stanford Functional Genomics Facility (SFGF) at Stanford University, on an Illumina NextSeq 550 instrument, using a high output kit, 1 × 76 cycles. Primer sequences and the sequencing construct layout are provided in Supplementary Data 7.

**Amplicon sequencing data analysis**. After bcl conversion and demultiplexing with Illumina bcl2fastq, the constant regions were trimmed using cutadapt[91]. The trimmed reads were aligned to the indexed reference of barcode sequences using Bowtie2 with the following options: -L 11 -N 0 --nofw[92]. The alignments were deduplicated based on UMIs using UMIcollapse[93] with -p 0.05 and counted using samtools idxstats. This pipeline yields a matrix of barcode read counts where rows are the different constructs in the library and columns are the different samples.

The count matrix was log transformed and normalized column-wise using a linear fit on the dilution series of spike-in constructs in each sample. For the calculation of RNA degradation coefficients in cells, we carried out a linear fit to log RNA abundance from the time course data, i.e., we fit an expression of $Y = \beta_0 + \beta_1 t$ where $Y$ is the normalized log RNA abundance and $t$ is the number of hours after transfection; $\beta_1$ is the degradation constant. The mRNA half-life was then derived as $ln(2)/\beta_1$. For the calculation of in solution degradation coefficients, sufficient data points were available to carry out a nonlinear fit directly to an exponential model, i.e., we fit an expression of $y = A \exp(-t/\tau)$, where $y$ is the fraction intact (RNA abundance normalized to initial abundance), $A$ is the amplitude, $t$ is the time of incubation in degradation buffer in hours, and $\tau$ is the degradation time constant. The mRNA half-life was then derived as $\tau ln(2)$. Time courses in which the observed fraction intact exceeded the fitted exponential by more than 0.05 in the last time point signaled RT-PCR amplification of misprimed non-full-length products and were filtered out of downstream analysis.

For polysome profiles, percent RNA abundances for each fraction were first calculated by scaling per-fraction values by the sum of all fractions. For the heatmap displays in the figures, column medians were also subtracted from each percent RNA value. For the calculation of ribosome load, the matrix of percent RNA abundances in fractions 4–9 (1–3 are free RNP fractions, and >9 have negligible abundance) were first multiplied by a weight vector representing the number of ribosomes in each fraction as determined by the A260 trace from the

fractionator, then the weighted abundances were summed across the row. For the calculation of polysome to monosome ratio, the sum of fractions 7–9 (>3 ribosomes) abundances were divided by fraction 4 (80S) abundance. For the calculation of monosome to 40S/60S ratio, fraction 4 (80S) abundance was divided by the sum of fraction 2 (40S/60S) abundance.

To calculate the expected protein levels assuming first order kinetics of mRNA translation and mRNA/protein decay, the following differential equations are used:

$$\frac{dM}{dt} = -k_m \cdot M(t) \quad (3)$$

$$\frac{dP}{dt} = k_t \cdot M(t) - k_p \cdot P(t) \quad (4)$$

where d$M$/d$t$ and d$P$/d$t$ are rates of change in mRNA and protein levels, respectively; $M(t)$ and $P(t)$ are moles of mRNA and protein at time $t$, respectively; $k_t$ is the translation rate constant; and $k_m$ and $k_p$ are rate constants of mRNA and protein decay, respectively. The analytical solution for $P(t)$ is proportional to:

$$P(t) \sim k_t \frac{e^{-k_p t} - e^{-k_m t}}{k_m - k_p} \quad (5)$$

where $m_0$ is the mass of mRNA present at $t = 0$, and $l$ is the mRNA length in nucleotides. $k_p$ is set to 0 since Nluc protein has negligible degradation as measured by luciferase activity in transiently Nluc-expressing HEK293 cells for at least 6 h after cycloheximide treatment, which allows assessment of protein degradation in the absence of further translation[94]. $k_m$ is the degradation constant obtained from the linear fit of in-cell time course RNA data ($-\beta_1$ above). $k_t$ is the ribosome load calculated by summing weighted RNA abundances from polysome profile data.

**SHAPE and DMS chemical mapping of full-length mRNAs**. For DMS-based chemical mapping of the LinearDesign-1 and Yellowstone mRNAs, 1 μg of RNA was brought to 10 μL in water, unfolded at 95 °C for 2 min, then snap-cooled on ice for 1 minute. The RNA was then mixed with 25 μL water and 10 μL 5X folding buffer (1.5 M sodium cacodylate pH 7.0, 50 mM MgCl$_2$) and folded at 37 °C for 30 min. The folded RNA was modified by adding 5 μL of 15% dimethyl sulfate (v/v in ethanol) or water (negative control). Both reactions were incubated at 37 °C for 6 min, quenched by the addition of 50 μL beta-mercaptoethanol, purified using the Zymo RNA Clean and Concentrator 5 kit (Zymo Research), and eluted in 12 μL water. For reverse transcription, 10 μL of the modified RNA was mixed with 1 μL of 10 μM oVT555, incubated at 65 °C for 5 min, and snap cooled on ice. Then, the template-primer mix was combined with 4 μL of 5X TGIRT First Strand Synthesis Buffer (250 mM Tris-HCl pH 8.3, 375 mM KCl, 15 mM MgCl$_2$), 2 μL 10 mM dNTPs, 1 μL freshly prepared 100 mM DTT, and 0.5 μL TGIRT-III (InGex LLC). The reaction was mixed and incubated at 57 °C for 3 h. RNA was then hydrolyzed by the addition of 10 μL hydrolysis buffer (0.5 M NaOH, 0.25 M EDTA) and incubation at 65 °C for 15 min. Hydrolysis was quenched by bringing the reaction volume to 50 μL with water, adding 100 μL of Oligo Binding Buffer (Zymo Research), and proceeding through the Zymo Oligo Clean and Concentrator (Zymo Research) purification protocol, eluting in 15 μL of water. Five microliters of the purified cDNA was amplified in a NEBNext Q5 HotStart master mix PCR reaction containing 0.5 μM each oVT554 and oVT555 with the following cycling conditions: 98 °C for 30 s, 10 cycles of 98 °C for 10 s followed by 72 °C for 60 s, with a final extension of 72 °C for 5 min. Products were purified using 0.9X Select-a-Size DNA Clean and Concentrator MagBeads (Zymo Research). Amplification of the full-length cDNA product was verified on an agarose gel stained with SYBRSafe (Invitrogen).

For SHAPE-based chemical mapping, 500 ng of RNA was brought to 12 μL in water and denatured at 95 °C for 2 min followed by snap-cooling on ice for 2 min. The RNA was then folded by adding 6 μL of 3.3X SHAPE folding buffer (333 mM HEPES pH 8.0, 333 mM NaCl, 33 mM MgCl$_2$) and incubating at 37 °C for 20 min. Nine microliters of the folded RNA was mixed with 1 μL of either 100 mM 1M7 (1-Methyl-7-nitroisatoic anhydride, freshly mixed in DMSO) or neat DMSO (negative control), mixed thoroughly by pipetting, and incubated at 37 °C for 75 s (roughly 5 1M7 hydrolysis half-lives). After chemical treatment, the volume of both reactions was brought to 50 μL with water, cleaned up with the Zymo RNA Clean and Concentrator 5 kit, and eluted in 12 μL of water. Ten microliters of eluted RNA was mixed with 1 μL of 200 ng/μL random nonamer primer (New England Biolabs), incubated at 65 °C for 5 min, then snap-cooled on ice. Then, 8 μL of 2.5X MaP buffer (125 mM Tris pH 8.0, 187.5 mM KCl, 25 mM DTT, 1.25 mM each dNTPs, 15 mM MnCl2), was added to the primer-template mixture, incubated at room temperature for 2 min, and then mixed with 1 μL of SuperScript II (Invitrogen). The reverse transcription reaction was thoroughly mixed by pipetting, incubated at room temperature for 10 min, then at 42 °C for 3 h. The reverse transcription enzyme was heat inactivated at 70 °C for 15 min, snap-cooled on ice, then immediately mixed with 8 μL NEBNext Second Strand Synthesis Reaction Buffer (New England Biolabs), 4 μL NEBNext Second Strand Synthesis Enzyme Mix (New England Biolabs), and 48 μL water. The second strand synthesis reaction was mixed thoroughly through pipetting and incubated in a thermocycler with the heated lid off at 16 °C for 60 min. The resulting double-stranded cDNA was purified with 1.8X volumes of Select-a-Size DNA Clean and Concentrator MagBeads (Zymo Research).

Double-stranded cDNA was prepared for Illumina sequencing with the NEBNext Ultra II FS DNA Library Prep Kit for Illumina (New England Biolabs) and iTru primers. Briefly, 100–500 ng of DNA from either 1M7 or DMS conditions in 26 µL of water was fragmented and end-repaired by adding 7 µL NEBNext Ultra II FS Reaction Buffer and 2 µL NEBNext Ultra II FS Enzyme Mix, mixed thoroughly by vortexing, and incubated in a thermocycler with heated lid on at 37 °C for 20 min, then 65 °C for 20 min. Fresh ligation adapter was prepared by denaturing a solution of 15 µM each of iTrusR1-stub and iTrusR2-stubRCp in salty TLE (10 mM Tris pH 8.0, 0.1 mM EDTA, 100 mM NaCl) at 95 °C for 1 min, then annealed via slow cooling at −0.1 °C/s to 25 °C. The fragmented and end-repaired DNA was mixed with 2.5 µL 15 µM ligation adapter, 30 µL NEBNext Ultra II FS Ligation Master Mix, and 1 µL NEBNext Ultra II FS Ligation Enhancer. The ligation reaction was incubated in a thermocycler (heated lid off) at 20 °C for 60 min. Ligated fragments were purified with 0.9X Select-a-Size DNA Clean and Concentrator MagBeads (Zymo Research) and eluted in 25 µL water. Dual-index sample barcodes were added through indexing PCR using the NEBNext Q5 Hot Start HiFi PCR Master Mix in a reaction with 0.5 µM each of iTru_5 and iTru_7 indexing primers with the following thermocycling parameters: 98 °C for 30 s, 8 cycles of 98 °C for 10 s, 55 °C for 10 s, 72 °C for 15 s, and a final extension at 72 °C for 5 min. The final sequencing libraries were purified with 0.9X volumes of Select-a-Size DNA Clean and Concentrator MagBeads (Zymo Research), quantified by qPCR using the iTaq Universal SYBR Green Supermix (Bio-Rad) with iTru_P5 and iTru_P7 primers, pooled in equimolar concentrations, and sequenced on an Illumina Miseq (Stanford Protein and Nucleic Acid Core Facility) for 600 cycles.

Demultiplexed reads were downloaded from the BaseSpace Sequence Hub (Illumina). Most analysis steps were carried out using the RNAFramework RNA structure probing analysis toolkit[95]. Using the 'rf-map' module, reads were trimmed with CutAdapt[91] and mapped to the wild-type RNA sequence using Bowtie 2[92]. Mutations were counted using the 'rf-count' module with '-m' flag, then normalized using the 'rf-norm' module with '-sm 4 -nm 2 -rb AC -nw 50 -dw' flags for DMS samples and '-sm 3 -nm 2 -rb N -nw 50 -dw' flags for 1M7 samples. These normalized reactivities (Supplementary Data 4) were plotted as-is, and also used to predict RNA secondary structure using the RNAStructure[96] 'Fold' command, implemented in Arnie (https://github.com/DasLab/arnie).

**High-throughput in-line and SHAPE probing on Eterna-designed RNA fragments (In-line-seq).** The In-line-seq experiments relied on a different pipeline for massively parallel RNA generation, treatment, and Illumina sequencing than mRNA experiments above. We describe these steps below; see ref. [97] for further details and Supplementary Data 7 for primers.

*Preparation of DNA templates.* DNA fragments encoding for RNA molecules from the Eterna 'OpenVaccine: Roll-your-own-structure' challenge were ordered in the form of a custom oligonucleotide pool of DNA (Custom Array/Genscript) with the 20-nt T7 RNA polymerase promoter sequence (5′-TTCTAATACGACTCACTATA-3′) prepended to each DNA. Amplification of the DNA template was performed via emulsion PCR. A hydrophobic solution ('oil phase') containing 80 µL of ABIL EM90 (Evonik Corporation), 1 µL of Triton X-100, and 1919 µL of mineral oil was vortexed for 5 min and then incubated on ice for 30 min. Then, 75 µL of water-soluble reaction mixture (liquid phase) containing 1X Phire Hot Start II buffer, 0.2 mM dNTPs, 1.5 µL of Phire II DNA polymerase, 2 µM of each primer (T7 promoter and Tail2 Reverse complement), 0.5 mg/ml of BSA, and 360 ng of the oligonucleotide pool was prepared. In a 1.0 ml glass vial (kept on ice and frozen at −20 °C overnight before use), 300 µL of the oil phase was added into the glass vial and vortexed at 1000 rpm for 5 min. Next, 10 µL of the liquid phase was added followed by 10 s of vortexing. This addition of the liquid phase followed by vortexing was repeated four times such that 50 µL of the liquid phase has been added to 300 µL of the oil phase in the vial. The now-mixed 350 µL of emulsion PCR solution was the subjected to the following thermocycling protocol: 98 °C for 30 s for initial denaturation, 42 cycles of amplification (98 °C for 10 s, 55 °C for 10 s, and 72 °C for 30 s), and final extension at 72 °C for 5 min.

The PCR was purified by adding 100 µL of mineral oil, followed by a brief vortex (~10 seconds) and centrifugation at 13,000 rcf for 10 min. The oil phase was then discarded. One microliter of diethyl ether was added, followed by a brief vortex (~10 s) and centrifugation at 13,000 rcf for 1 minute. The upper layer (termed the detergent layer) was then discarded. Extraction with diethyl ether was repeated three times to ensure thorough purification. The resulting product was then incubated at 37 °C for 5 min, adjusted to a final volume of 40 µL with nuclease-free water, then purified with 72 µL of Ampure bead XP (Beckman Coulter) following the standard protocol specified by the vendor. Finally, DNA was eluted into 20 µL of nuclease-free water.

*Preparation of RNA templates.* A library of RNA molecules was then prepared from the amplified DNA template using the TranscriptAid T7 High Yield Transcription Kit (Thermofisher, K0441) using the reaction mixture specified by the vendor. Transcription was performed with incubation at 37 °C for 3 h. After transcription, the DNA template was removed through the addition of 2 µL of DNAse I (add vendor) followed by incubation at 37 °C for 30 min. After DNA digestion, the RNA was purified using a mixture of AMPure XP beads (Beckman Coulter) with 40%

polyethylene glycol (mixed in a 7:3 ratio). Final elution into 25 µL of nuclease-free water yielded purified RNA.

*Degradation in-line probing of RNA samples.* For degradation experiments, 45–50 pmol of RNA was subject to four conditions: (1) 50 mM Na-CHES buffer (pH 10.0) at room temperature without added MgCl₂; (2) 50 mM Na-CHES buffer (pH 10.0) at room temperature with 10 mM MgCl₂; (3) phosphate buffered saline (PBS, pH 7.2; Thermo Fisher Scientific-Gibco 20012027) at 50 °C without added MgCl₂; and (4) PBS (pH 7.2) at 50 °C with 10 mM MgCl₂.

For degradation reactions containing MgCl₂, RNA was collected at 0 and 24 h time points. For reactions without MgCl₂, timepoints were collected at 0 and 7 days. At each timepoint, the degradation reaction was quenched with 15 µL of 500 mM of Tris-HCl (pH 7) and 3 µL of 500 mM of Na-EDTA. Quenched samples were brought to a final volume of 100 µL with nuclease-free water. The RNA was purified through precipitation as follows. 1.5 µL of Glyco Blue (Thermo Fisher), 10 µL of 3 M sodium acetate (pH 5.2), and 330 µL of cold 100% ethanol were added, the reaction mixture mixed, and incubated on dry ice for 20–30 min. After incubation, the reaction was centrifuged at 21,000 rcf for 30 min. The resulting pellet was washed twice with 500 µL of cold 70% ethanol, and pelleted after each wash step. Finally, the reaction was dried for 10 min at room temperature to remove any residual ethanol, and resuspended in 5 µL of nuclease-free water.

*Structure probing of RNA samples.* In parallel with the in-line hydrolytic degradation conditions above, we carried out SHAPE structure probing experiments[65], as follows. 15 pmol of purified RNA was added to 2 µL of 500 mM HEPES buffer (pH 8.0) and denatured at 90 °C for 3 min. The reaction was then cooled down to room temperature over 10 min. Two microliters of 100 mM MgCl₂ was then added, followed by incubation at 50 °C for 30 min. The sample was cooled down to room temperature over 20 min before addition of 5 µL of 1-methyl-7-nitroisatoic anhydride (1M7, 8.48 mg/mL of DMSO) followed by incubation at room temperature for 15 min, and brought to a final volume of 20 µL with nuclease-free water. The reaction was quenched with 5 µL of 500 mM Na-MES pH 6.0, the reaction was adjusted to be 100 µL, and purified with ethanol precipitation as above. As a control, a sample was prepared in parallel without the addition of 1M7 but subject to the same protocol described above.

*Preparation of cDNA and sequencing.* cDNA was prepared from the six RNA samples (two from structure probing, four from degradation). Five microliters of purified RNA was added to a reaction mixture containing 1x First Strand buffer (Thermo Fisher), 5 mM dithiothreitol (DTT), 0.8 mM dNTPs, 0.6 µL of Super-Script III RTase (Thermo Fisher) to a final volume of 15 µL. The reaction was incubated at 48 °C for 40 min and stopped with 5 µL of 0.4 M sodium hydroxide. The reaction was then incubated at 90 C for 3 minutes, cooled on ice for 3 min, and neutralized with 2 µL of quench mix (prepared as 2 mL of 5 M sodium chloride, 3 mL of 3 M sodium acetate, 2 mL of 2 M hydrochloric acid).

cDNA was pooled down with 1.5 µL of Oligo C' beads[65] (in house magnetic beads prepared by immobilizing 2x Biotin oligonucleotides with Dynabeads™ MyOne™ Streptavidin C1; Thermo Fisher Scientific 65001), washed twice with 70% ethanol, then resuspended in 3.0 µL of water. We pooled 1.5 µL of each sample together, and took 9 µL of cDNA to continue to ligation an Illumina adapter by using Circ. Ligase I (Lucigen) at 68 °C for 2 h. The reaction was stopped by incubation at 80 °C for 10 min. cDNA was added to 10 µl of 5 M NaCl and pulled down with a magnetic stand and washed with 70% ethanol; the ligated product was resuspended in 15 µL H₂O.

The ligated product was quantified by qPCR. dsDNA at 3 nM concentration was sequenced using an Illumina Miseq (High output, Read 1 = 101 cycles and Read 2 = 51 cycles). The resulting data were analyzed using MAPseeker (https://ribokit.github.io/MAPseeker)[98] following the recommended steps for sequence assignment, peak fitting, background subtraction of the no-modification control, correction for signal attenuation, and reactivity profile normalization.

**In-line and SHAPE probing by capillary electrophoresis.** One-by-one follow-up to profile degradation of RNA fragments at single-nucleotide resolution through capillary electrophoresis was carried out with some differences to the pooled In-line-seq experiments above. We describe these steps below; see ref. [97] for further details and Supplementary Data 7 for primers.

*DNA template preparation.* DNA templates were designed to include the 20-nt T7 RNA polymerase promoter sequence (5′-TTCTAATACGACTCACTATA-3′) followed by the remaining sequence encoding desired RNA. Double-stranded templates were prepared by extension of 60-nt DNA oligomers (IDT, Integrated DNA Technologies) with Phusion DNA polymerase (Finnzymes), using the following thermocycler protocol: denaturation for 30 s at 98 °C, 35 cycles of denaturation for 10 s at 98 °C annealing for 30 s at 60–64 °C, extension for 30 s at 72 °C; final extension for 10 min at 72 °C and cooling to 4 °C.

DNA samples were purified with AMPure magnetic beads (Agencourt, Beckman Coulter) following the manufacturer's instructions. Sample concentrations were estimated based on UV absorbance at 260 nm measured on Nanodrop 100 or 8000 spectrophotometers. Verification of template length was

accomplished by electrophoresis of all samples and 10-bp and 20-bp ladder length standards (Thermo Scientific O'RangeRuler SM1313 & SM1323) in 4% agarose gels (containing 0.5 mg/mL ethidium bromide) and 1x TBE (100 mM Tris, 83 mM boric acid, 1 mM disodium EDTA). All sample manipulations, including the following steps, were carried out in 96-well V-shaped polypropylene microplates (Greiner).

*Preparation of RNA templates.* In vitro transcription reactions were carried out in 40 μL volumes with 10 pmol of DNA template, using the TranscriptAid T7 High Yield Transcription Kit (Thermo Fisher, K0441). In some syntheses, pseudouridine-5′-triphosphate (TriLink Biotechnologies, N-1019) or N1-methylpseudouridine-5′-triphosphate (TriLink Biotechnologies, N-1081) were used to replace regular UTP. Reactions were incubated for 3 h at 37 °C, followed by degradation of DNA template with 2 μL of DNase I at 37 °C for 30 min. RNA samples were purified with 1.8x volume of AMPure XP beads (Beckman Coulter) mixed with 40% PEG-8000 (ratio of 7:3), following the manufacturer's instructions. Concentrations were measured by absorbance at 260 nm on Nanodrop 100 or 8000 spectrophotometers.

*In-line probing (hydrolytic degradation profiling).* For degradation experiments, 750 ng of RNA was subjected to 50 mM Na-CHES buffer (pH 10) at room temperature with 10 mM MgCl₂ at a final volume of 30 μL. RNA was collected at 0- and 24-h time points. At each timepoint, 15 μL of reaction was taken, and degradation quenched with 7.5 μL of 500 mM of Tris-HCl (pH 7) and 1.5 μL of 500 mM of Na-EDTA. Quenched samples were purified with 2 μL of Oligo dT bead (Thermo Fisher, AM1922) and 0.8 μL of 10 μM FAM-A20-Tail2. RNA, bead, and primer were incubated at room temperature for 15 min, pulled down by magnetic stand for 10 min, and washed twice with 70% ethanol and left to dry on the bead. RNA was eluted in 2.5 μL of nuclease-free water (RNA was bound with OligodT bead and FAM.A20 Tail2 at this point). Subsequently, these RNAs were further treated and purified with steps analogous to SHAPE probing (see next) to enable side-by-side comparisons. RNAs were added to 2 μL of 500 mM Na-HEPES buffer (pH 8.0) and denatured at 90 °C for 3 min. The reaction was then cooled down to room temperature over 10 min. Two microliters of 100 mM MgCl₂ was then added, followed by incubation at 50 °C for 30 min. The sample was cooled down to room temperature over 20 min before the addition of 5 μL of nuclease-free water, followed by incubation at room temperature for 15 min, and brought to a final volume of 20 μL with nuclease-free water. The RNA sample was further purified by incubating the sample with 5.0 μL of Na-MES, pH 6.0, 3.0 μL of 5 M NaCl, and brought to a final volume of 10 μL with nuclease-free water. The reaction mixture was incubated at room temp for 15 min, pulled down by 96-post magnetic stand for 10 min, washed twice with 70% ethanol, and allowed to dry, before adding 2.5 μL of nuclease-free water.

*SHAPE experiments.* For SHAPE structure probing experiments, 1.2 pmol of purified RNA was added to 2 μL of 500 mM Na- HEPES buffer (pH 8.0) and denatured at 90 °C for 3 min. The reaction was then cooled down to room temperature over 10 min. 2 μL of 100 mM MgCl₂ was then added, followed by incubation at 50 °C for 30 min. The sample was cooled down to room temperature over 20 min before addition of 5 μL of nuclease-free water (negative control) or 1-methyl-7-nitroisatoic anhydride (1M7, 8.48 mg/mL of DMSO) followed by incubation at room temperature for 15 min, and brought to a final volume of 20 μL with nuclease-free water. The SHAPE-RNA sample was further purified by incubating the sample with 5.0 μL of Na-MES, pH 6.0, 3.0 μL of 5 M NaCl, 1.5 μL of Oligo dT bead, 0.25 μL of 10 μM FAM-A20-Tail2, and brought to a final volume of 10 μL with nuclease-free water. The reaction mixture was incubated at room temp for 15 min, pulled down by 96-post magnetic stand for 10 min, washed twice with 70% ethanol and allowed to dry, before adding 2.5 μL of nuclease-free water.

*Preparation of in-line probing and SHAPE samples for capillary electrophoresis.* cDNA was prepared from in-line probing and SHAPE RNA samples as follows (note that above procedures leave RNA bound to FAM-A20-Tail2 reverse transcription primers which are in turn bound to Oligo dT beads). 2.5 μL of purified RNA was added to a reaction mixture containing 1x First Strand buffer (Thermo Fisher), 5 mM dithiothreitol (DTT), 0.8 mM dNTPs, 0.2 μL of SS-III RTase (Thermo Fisher) to a final volume of 5.0 μL. The reaction was incubated at 48 °C for 40 min, and stopped with 5 μL of 0.4 M sodium hydroxide. The reaction was then incubated at 90 °C for 3 minutes, cooled on ice for 3 min, and neutralized with 2 μL of quench mix (2 mL of 5 M sodium chloride, 3 mL of 3 M sodium acetate, 2 mL of 2 M hydrochloric acid). For four cDNA reference ladders, each of four ddNTPs (GE Healthcare 27-2045-01) with a ddNTP/dNTP ratio of 1.25 (0.1 mM/ 0.08 mM) was used in the reverse-transcription reaction.

cDNA was pulled down on a 96-post magnetic stand and washed two times with 100 μL 70% ethanol. To elute the bound cDNA, the magnetic beads were resuspended in 10.0625 μL ROX350 (Thermo Fisher Scientific 401735)/Hi-Di (0.0625 μL of ROX 350 ladder in 10 μL of Hi-Di formamide) and incubated at room temperature for 20 min. The cDNA was further diluted by 1/3 and 1/10 in ROX350/HiDi and samples loaded onto capillary electrophoresis sequencers (ABI-3730) on capillary electrophoresis (CE) services rendered by ELIM Biopharmaceuticals. CE data were analyzed using the HiTRACE 2.0 package[99]

(https://github.com/ribokit/HiTRACE), following the recommended steps for sequence assignment, peak fitting, background subtraction of the no-modification control, correction for signal attenuation, and reactivity profile normalization.

**Measurement of in-solution mRNA stability by capillary electrophoresis.** For one-by-one measurement of in-solution mRNA stability, in vitro transcribed mRNA was incubated in a degradation buffer over ten time points (0, 0.5, 1.0, 1.5, 2, 3, 4, 5, 18, and 24 h), then analyzed by capillary electrophoresis.

For each time point, 1.6 pmol of mRNA brought to 10 μL in a buffer containing 50 mM Na-CHES at pH 10 with 10 mM MgCl₂, and the reaction was incubated at 25 °C. When the incubation period was reached for each time point, 5 μL of Tris-HCl at pH 7 and 1 μL of 500 mM EDTA in nuclease free water was added to quench the degradation reaction, and frozen for further analysis. After the final time point (24 h), 4 μL of each mRNA degradation sample (out of a total stored volume of 16 μL) was taken, and mixed with 1 μL of a control RNA at a concentration of 50 ng/μL. For these experiments, we opted to use the P4-P6 domain of the *Tetrahymena* ribozyme with two added hairpins[97] (~239 nt) as our control. The RNA mixture was then purified using a mixture of AMPure XP beads (Beckman Coulter) with 40% polyethylene glycol (mixed in a 7:3 ratio). The resulting RNA was eluted into 4.5 μL of RNAse-free water for analysis on the 2100 Bioanalyzer (Agilent) using the RNA-Nano Eukaryote protocol.

The data from the Bioanalyzer were analyzed using a custom script that performs the following analysis. We first converted elution times to nucleotides based on a ladder control (25, 200, 500, 1000, 2000, and 4000 nts). We then estimated relative mRNA amounts based on peak areas at expected band lengths (in our case, ~900 nucleotides for the mRNAs of interest and ~265 nucleotides for the control). When calculating peak areas, we performed background subtraction, where the background was defined as the area under a linear line in the range of nucleotides used for the peak area. Normalization was performed using two different methods used to cross-validate. First, we normalized the peak areas of full-length mRNA to the control P4-P6 domain RNA that was spiked into the samples after degradation was performed. Second, we also normalized peak areas of full-length mRNAs to the total amount of RNA in the lane less the peak area of the bands of interest (between ~20 and 1000 nucleotides in our case), assuming that the majority of the other RNA in the lane were degradation products from the mRNA of interest. These distinct approaches to normalizing the data gave the same results for half-life relative to reference mRNA within estimated error (see below). After calculations of normalized peak areas, we then calculated fraction intact values for each mRNA by dividing the normalized area across the ten timepoints by the normalized area at the start (0 h).

$$Fraction\ Intact_i = \frac{Normalized\ Area_i}{Normalized\ Area_{0hours}} \quad (6)$$

For each sample, we fit fraction intact values across the different timepoints to an exponential function,

$$F_i = Ae^{-\tau/t} \quad (7)$$

where $F_i$ is an array of fraction intact values across multiple time points, $A$ is the amplitude of the exponential decay function, $\tau$ is the time constant, and $t$ is an array of time points in hours. We then used the time constant to calculate the in vitro half-life of mRNA:

$$Half - life = \ln(2)\tau \quad (8)$$

Scripts and data are available at https://github.com/DasLab/openvaccine-CE-analysis.

**Polysome selection and library preparation.** The variant 5′ UTR is composed of: fixed first 29 nt of *hHBB*, variable 35 nt (initially degenerate) and 6 nt Kozak consensus. See Supplementary Data 7 for the detailed construct layout. To generate the reporter mRNA pool containing the variant 5′ UTR library, IVT template was first assembled by PCR under the following conditions: 4 μL 10x AccuPrime Pfx Reaction Mix, 0.4 pmol HBB29_N35 amplicon, 0.4 pmol Nluc_HBB_3UTR, 0.4 μL AccuPrime Pfx Polymerase in 40 μL of total reaction volume. Cycling conditions are: 95 °C for 120 s, and 19 cycles of 95 °C for 15 s, 66 °C for 30 s, 68 °C for 75 s. PCR product was purified on silica columns (NEB T1034) and amplified with under the following conditions: 4 μL 10X AccuPrime Pfx Reaction Mix, 4 μL 10 μM T7_28_HBB_30_F, 4 μL 10 μM Nanoluc_ORF_R, 0.4 μL AccuPrime Pfx Polymerase in 40 μL total reaction volume. Cycling conditions are: 95 °C for 120 s, and 4 cycles of 95 °C for 15 s, 66 °C for 30 s, 68 °C for 75 s. The mRNA was in vitro transcribed, capped and polyadenylated as described above. This yields an estimated initial starting degenerate pool complexity of ~2.4 × 10¹¹.

Transfection of HEK-293 cells and sucrose gradient fractionation were performed as described above. Equal volumes of fractions 10–16 were pooled and RNA was by acidic phenol chloroform extraction followed by column purification (Zymo Research, R1013) as described above. 1/3 lysate volume was kept as input before layering onto the sucrose gradient and RNA was extracted from the input lysate by Trizol extraction followed by column purification. 1.5 μg RNA in 5.5 μL was mixed with 0.5 μL 2uM RT_Nluc26_UMI12_Read1Partial (Supplementary Data 7) and 0.5 μL 10 mM dNTPs each. The RNA samples were then denatured at 65 °C for 5 min and chilled to 4 °C. 3.5 μL reverse transcription mix was added to

10 μL total reaction volume: 2 μL 5x Superscript IV buffer, 0.5 μL 10 mM DTT, 0.5 μL Superase-In (ThermoFisher Scientific, AM2694), 0.5 μL Superscript IV (Thermo 18091050). The reaction was incubated at 55 °C for 45 min and inactivated at 80 °C for 10 min. Variant 5′ UTR amplicon was amplified from the reverse transcription reaction via PCR under the following reaction conditions: 4 μL RT reaction, 40 μL 2x Q5 Hot Start Master Mix (NEB M0494S), 0.8 μL 100x SYBR (Thermo S7563), 4 μL 10 μM T7_28_HBB_29_F, 4 μL 10 μM Nanoluc_ORF_R, in 80 μL total reaction volume. Cycling conditions were as follows: 98 °C for 60 s, and 15 cycles of 98 °C for 10 s, 68 °C for 10 s, 72 °C for 10 s. PCR product was purified on silica columns (NEB T1034) and assembly with Nluc_HBB_3UTR fragment was performed as described above for initial preparation of IVT template using HBB29_N35 amplicon. The mRNA was in vitro transcribed, capped, and polyadenylated as described above. The same process of transfection, fractionation, reverse transcription, PCR amplification, assembly and in vitro transcription was repeated.

For sequencing library preparation, the RT reaction was PCR amplified under the following conditions: 1 μL RT reaction, 10 μL 2x Q5 Hot Start Master Mix (NEB M0494S), 0.2 μL 100x SYBR (Thermo S7563), 1 μL 10 μM Read1, 1 μL 10 μM Read2Partial_HBB29 in 20 μL total reaction volume. Cycling conditions were as follows: 98 °C for 60 s, and 15 cycles of 98 °C for 10 s, 68 °C for 10 s, 72 °C for 10 s. Sequencing adaptors were added using the following conditions for final round PCR: 1 μL first round PCR reaction, 10 μL 2x Q5 Hot Start Master Mix, 0.2 μL 100x SYBR, 1 μL 10 μM NEBNext Index Primer (NEB E7335, NEB E7500, NEB E7710, NEB E7730, NEB E6609), 1 μL 10 μM NEBNext Universal PCR Primer in 20 μL total volume. Cycling conditions are: 98 °C for 60 s, and 5 cycles of 98 °C for 10 s, 72 °C for 10 s. All barcoded samples were then pooled at equal volumes and purified with 1.1x SPRIselect beads Beckman Coulter B23317). Sequencing was performed at the Stanford Functional Genomics Facility (SFGF) at Stanford University, on the Illumina NextSeq 550 instrument, using a high output kit, $1 \times 81$ cycles. Primer sequences and the sequencing construct layout are provided in Supplementary Data 7.

**Polysome selection library sequencing data analysis**. Following adapter trimming, 670,440 sequences with at least 10 summed read count across all libraries combined were set as the reference. Each library was aligned to this indexed reference using Bowtie2[92]. Only uniquely mapping reads with edit distance ≤3 were retained. Alignments were further deduplicated using UMIcollapse (-p 0.05, -k 1)[93]. This results in the matrix of read count where rows are different sequence variants and columns are the samples.

Normalized counts were obtained by dividing the matrix column-wise by total read counts per sample. For sequence variants with at least 15 reads in any one of the samples, a regression model was fitted on normalized read counts with the sequential selection rounds as ordinal predictors, penalizing differences between coefficients of adjacent groups (R package ordPens)[100]. False discovery rate was estimated by Benjamini-Hochberg procedure. For choosing the final set of candidates, the criteria of ≥15 read counts in the final round polysome selection library and ≥2 fold enrichment over input in the final round was also required.

For analysis of k-mers, 1 million reads (prior to alignment and analysis for highly enriched individual sequences) were sampled from each library. Position-specific k-mers were counted and statistical significance of pair-wise enrichments for each position-specific k-mers are calculated using kpLogo[101]. The parameters were: zero-order Markov model background; $2 \leq k \leq 6$; -shift 0 and -max-shift 0; binomial test with Bonferroni correction.

**Design algorithms/analysis**. The protein sequences for each target were used to generate DNA sequences at Integrated DNA Technologies (IDT, https://www.idtdna.com/CodonOpt), Twist Biosciences (https://ecommerce.twistdna.com/app), and GENEWIZ (https://clims4.genewiz.com/Toolbox/CodonOptimization). Solutions from LinearDesign were obtained using the LinearDesign server (http://rna.baidu.com/). Ribotree solutions were terminated after 6000 iterations. The RiboTree code is available for noncommercial use at https://eternagame.org/about/software, with example usage to replicate runs for this work provided.

Structure prediction and ensemble-based calculations were performed using LinearFold and LinearPartition with ViennaRNA, CONTRAfold, and EternaFold parameters. Secondary structure features were calculated from predicted MFE structures using RiboGraphViz (www.github.com/DasLab/RiboGraphViz). CAI was calculated as the geometric mean of the relative usage frequency of codons along the length of the coding region, as previously reported[46] (9):

$$CAI = \left(\prod_{i=2}^{L} w_i\right)^{L-1}; w_i = \frac{f_i}{\max(f_j)} \tag{9}$$

where $f_j$ represents the frequency of all codons coding for amino acid at position i. Scripts to reproduce degradation and structure predictions are available in the "OpenVaccine-solves" database under an Open COVID license at https://eternagame.org/about/software.

**DegScore Linear Regression model**. A Ridge regression model in scikit-learn (https://scikit-learn.org/) was used to create the DegScore linear regression model, available at www.github.com/eternagame/DegScore.

For the 24 Nluc constructs synthesized with pseudouridine (PSU), we wished to test if accounting for increased stability of PSU resulted in improved performance. Without accounting for PSU, we found that the DegScore predictions had a correlation of (Spearman $R = 0.55$, $p < 0.001$) for the 24 constructs. We modified DegScore to set the prediction per nucleotide at each Uridine to be zero, to mimic minimal contributions to degradation. This simple change to the DegScore model resulted in a notable improvement in correlation (Spearman $R = 0.67$, $p < 0.001$, Fig. 4e), and was used in analysis for constructs synthesized with PSU.

**Statistics and reproducibility**. In all figures, data are presented as mean, SD or SEM as stated in the figure legends, and *$p \leq 0.05$ is considered significant (ns: $p > 0.05$; *$p \leq 0.05$; **$p \leq 0.01$; ***$p \leq 0.001$; ****$p \leq 0.0001$). Blinding and randomization were not used in any of the experiments. Number of independent biological replicates used for the experiments are listed in the figure legends. Tests, two-tailed unpaired Student's t-test if not stated otherwise, and specific p-values used are indicated in the figure legends. In all cases, multiple independent experiments were performed on different days to verify the reproducibility of experimental findings. Errors for in-cell degradation coefficients are standard errors of coefficient estimates. Errors for in-cell ribosome load are estimated by bootstrapping where fraction labels are shuffled and before scaling by spike-in normalization factors. Errors for predicted expression values are estimated by Taylor series method. Error and significance tests for single-exponential in-solution half-lives were estimated by non-parametric bootstrapping, except for data on the 24 nanoluciferase measurements in Fig. 4, which made use of error estimates derived from triplicate measurements.

**Reporting summary**. Further information on research design is available in the Nature Research Reporting Summary linked to this article.

## Data availability

The data that support this study are available from the corresponding authors upon reasonable request. Raw sequencing data from PERSIST-seq and In-line-seq experiments are available at the Gene Expression Omnibus (GEO) at accession number GSE173083. Single-nucleotide-resolution in-line probing and SHAPE data are deposited at the following RNA Mapping Database[102] (http://rmdb.stanford.edu) accession numbers: RYOS1_NMD_0000 (no modification); RYOS1_MGPH_0000 (10 mM Mg$^{2+}$, pH 10, 24 °C, 1 day). RYOS1_PH10_0000 (0 mM Mg$^{2+}$, pH 10, 24 °C, 7 days); RYOS1_MG50_0000 (10 mM Mg$^{2+}$, pH 7.2, 50 °C, 1 day); RYOS1_50C_0000 (0 mM Mg$^{2+}$, pH 7.2, 50 °C, 7 days); SHAPE_RYOS_0620 (SHAPE 1M7 reactivity). One-by-one capillary electrophoresis follow-ups are available in the RNA Mapping Database (RMDB) with the accession codes: RYOSFL_MOD_0001, RYOSFL_MOD_0002, RYOSFL_MOD_0003, RYOSFL_MOD_0004, RYOSFL_MOD_0005, RYOSFL_MOD_0006, RYOSFL_MOD_0007, RYOSFL_MOD_0008. Source data are provided with this paper.

## Code availability

Scripts and data for the measurement of in-solution mRNA stability by capillary electrophoresis are available at https://github.com/DasLab/openvaccine-CE-analysis. Code and formatted datasets to reproduce the linear DegScore model and degradation prediction calculations is available at https://github.com/eternagame/DegScore. PERSIST-seq processing pipeline is available at https://github.com/barnalab/persist.

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

## Acknowledgements

We thank the Barna and Das lab members for constructive criticism critical for the research. We thank Ramya Rangan (Stanford) for advice on SARS-CoV-2 structure; Ivan N. Zheludev (Stanford) for performing pilot experiments; Kelsey L. Hickey and Jonathan S. Weissman (UCSF) and Conor J. Howard (UCSF) for kindly sharing plasmids; John Coller and Dhananjay Wagh (Stanford Functional Genomics Facility (SFGF)), and Michael Eckart, Kyle Fukui, Jennifer Okamoto, Lisa Sharp, Camilla Kao, and Jessica Corkern for Stanford research support; Liang Huang (Oregon State University, Baidu Research USA) for discussions of LinearDesign; and Kari Efferen (Pfizer) for valuable feedback. We thank Howard Y. Chang (Stanford University); Hani Choudhry (King Abdullaziz University); Anthony Goldbloom, Maggie Demkin, and Walter Reade (Kaggle, Inc.); and Dani Braun, Austin Chen, Sharif Ezzat, Abhi Garg, Johannes Häggqvist, Tamas Kalman, Kevin Lin, and Amine Rehioui for advice on and development of Eterna OpenVaccine challenges. We thank Helga Leppek for providing handmade masks for essential lab research. Eterna participant usernames and full names, when volunteered, are provided in Supplementary Data 8. RiboTree calculations were performed on the Stanford Sherlock cluster. This work was supported by NIH grants R01HD086634 (M.B.), R21 CA219847 (R.D.), and R35 GM122579 (R.D.), the National Science Foundation GRFP (H.K.W.S., D.S.K.), a Benchmark Stanford Graduate Fellowship (G.W.B, C.A.C.), a Stanford ChEM-H "Stanford RISE" seed grant (H.K.W.S.), a Canadian Institutes of Health Research Postdoctoral Fellowship (C.K.), Paul and Daisy Soros Fellowships for New Americans (G.C.T), Stanford Medical Scientist Training Program (A.X., G.C.T.), a Human Frontier Science Program Fellowship (K.F.), and a sponsored research award (PE_IC2020-0726 from Pfizer, Inc to R.D., M.B.). D.R. is supported by an EMBO Long-term Fellowship (ALTF 1042-2019) and a Human Frontier Science Program Fellowship (LT000218/2020-L). A.X. is supported by a Stanford Bio-X/ SIGF fellowship and NIH fellowship 1F30HD100123. K.L. is supported by an EMBO Long-Term Fellowship (ALTF 539-2015), is the Layton Family Fellow of the Damon Runyon Cancer Research Foundation (DRG-2237-15), and is supported by the Katharine McCormick Advanced Postdoctoral Scholar Fellowship to Support Women in Academic Medicine (2019). M.B. is a New York Stem Cell Foundation Robertson Investigator. Further financial support came from gifts to the Eterna OpenVaccine project from donors listed in Supplementary Data 9.

## Author contributions

A.F.X. and D.S.K. contributed equally to this work. M.B. and R.D. conceived the project; M.B., R.D., K.L., G.W.B., H.K.W-S., D.S.K., W.K., and C.H.K. designed the experiments; K.L., G.W.B., W.K., H.K.W-S., C.H.K., A.F.X., V.V.T., G.C.T., K.F., F.D., H.C., P.G., J.W., F.M., S.S., and E.S. performed the experiments; G.W.B., H.K.W-S., V.V.T., C.C., D.R., R.D., D.S.K., and E.P. performed the bioinformatic and statistical analysis. R.W-O., B.T., J.J.N., J.R., H.K.W.-S., R.D., A.M.W., and D.S.K. designed and coordinated Eterna challenges. P.R.D. and A.S. led experiments performed at Pfizer. All authors contributed to analysis of the data, interpreted the results, and wrote the paper. All authors read and approved the manuscript.

## Competing interests

Stanford University has submitted provisional patent applications related to use of the *Hoxa9* P4 stem-loop (K.L. and M.B.), and the SARS-CoV2 5′ UTR (K.L. and M.B.), computational design of mRNAs (H.K.W.-S., D.S.K., C.C., E.S., R.D.), chemically modified nucleotides to stabilize RNA therapeutics (W.K. and R.D.), and design of reporter mRNAs and the PERSIST-seq platform (M.B., G.W.B., and K.L.). R.D., D.S.K., W.K., and C.H.K. have initiated a commercial venture focused on improving RNA design. R.D. received an honorarium for speaking at Pfizer's 2021 mRNA science day. F.D., H.C., P.G., J.W., F.M., S.S., A.S., and P.R.D. are or were employees of Pfizer and may hold stock options. P.R.D. is currently an employee and shareholder of GlaxoSmithKline (GSK).
