## [Peer Review File · Nature Communications]

Combinatorial optimization of mRNA structure, stability, and translation for RNA-based therapeuticsReviewers' Comments:

Reviewer #1:

Remarks to the Author:

In this study, Leppek et al reported an RNA sequencing-based platform to measure and predict in-cell mRNA stability, translation efficiency, and in-solution mRNA stability. The authors selected a group of synthetic mRNAs (a total of 223) with different combination of 5'UTR, CDS, and 3'UTR. The major goal is to establish some "rules" guiding mRNA design to achieve optimal stability and translation efficiency.

Therapeutic mRNA engineering is of significance given the recent success of mRNA vaccines targeting coronavirus. From RNA structures, ribosome loading, stability, and modification, the authors employed comprehensive approaches in order to establish "superfolder" mRNAs. Unfortunately, after finishing reading of the manuscript, this reader is totally lost. What exactly is the "superfolder" mRNAs that hold therapeutic potential? More puzzling, some of the conclusions are contradictory to the conventional view. For instance, how could structured CDS improve in-cell translation efficiency? How could polysome-associated mRNAs are less stable? The entire manuscript also suffers from many overstatements. Overall, this manuscript does not seem to deliver the useful information as it is supposed to do.

Main concerns

1. The entire library contains only 223 mRNAs, which is far from "the first large-scale screen of mRNA redesigned across its entire length". Given the large variations in 5'UTR, CDS, and 3'UTR, how robust is the statistical power?
2. Figure 1E, it is quite surprising to find that the commonly used beta-actin (hACTB) 5'UTR is the poorest in ribosome loading. By contrast, the scramble 5'UTR (scrUTR) behaved much better. The authors did not even discuss about this.
3. Figure 1 shows GC-rich CDS has almost the highest ribosome loading. In human cells, it has been believed that increasing GC-content can improve protein output because GC-rich codons tend to be optimal for translation in human cells. Therefore, a lower ribosome density is expected in GC-rich CDS, since elongating ribosomes move faster on these mRNAs. This is quite opposite to ribosome loading reported in Figure 1.
4. Supplemental Figure 2. The massively paralleled assay based on the 35 random nucleotide sequence in 5'UTR is expected to generate some exciting "principles" in optimal mRNA design. Oddly, these results were described in a brief manner.
5. Figure 2 shows that mRNAs with higher polysome/monosome ratio tends to have shorter half-life. Does that suggest translating ribosome trigger RNA degradation? Without considering other factors, e.g. sequence length or different sequence composition (e.g. different GC-content), it is not convincing to make a conclusion that polysome mRNAs are short half-life mRNAs. In fact, from the right panel of Figure 2B, only a small subset points (mRNAs) shows strongly negative correlation of half-life to polysome/monosome. They may use their random sequence reporter (Supplemental Figure 2) to further investigate such correlation.
6. Figure 3F, it is unclear how the authors conclude that "substitution of U with ψ or m1 ψ led to remarkable suppression of in-line hydrolysis". Is the 1.2 fold increase (Figure 3G and 3H) statistically significant?
7. Figure 4A, the translational output assay is probably the most useful in judging the mRNA potency. However, the luciferase activity (especially from 24 hr) is largely inconsistent with the previous design "principles".
8. For the therapeutic point of view, how to achieve "CDS" design? The coding region does not have lots of room for engineering, except changing codon optimality. On the other words, how to create "superfolder" mRNAs by changing CDS sequences? This is utterly confusing.

Reviewer #2:

Remarks to the Author:

The manuscript by Leppek et al uses different strategies to optimize the translatability and stability of mRNAs inside cells and in solution in order for maximum protein production for RNA vaccines. They developed a strategy to test ribosome load and decay rates of their RNAs and tested different 5' and 3'UTRs as well as different CDS features using cloud sourcing. They observed that decay is more important than ribosome load for long term translation. They also identified structural features in the CDS and showed that replacing U with pseudouridine increases protein production. Collectively, they were able to improve the protein production of their target RNAs using their design rules. This paper comes at an important time whereby RNA vaccines show great promise in our combat against SARS-CoV-2. However, current versions of RNA vaccines still suffer from degradation issues and would benefit from better design in terms of stability and translatability. I have some major concerns that needs to be addressed before this manuscript should be published.

1. The authors use reverse transcription, PCR and sequencing to identify full length transcripts that have not undegone in cell or in solution decay. They conclude that the lengths of CDS make a difference to stability. How do the authors know that the differences in decay is not due to increased RT stoppages (dropoff) along the transcript because of longer or more structured CDS/UTRs?
2. From their screens, Nluc with 5' and 3'den2 UTRs have high ribosome load. Why was the truncation and optimization done on SARS-CoV-2 UTRs and not den2 5'and 3'UTRs? In addition, den2 5'UTR data is not illustrated in Figure 2E.
3. What does a higher ribosome load of 1.7-2.3 mean? How many fold increase in translation does that correspond to?
4. With regards to the library of randomized 5'UTR sequences, how long are the 5'UTRs and how complex is the library?
5. How does the results of lineardesign mRNA structure optimization compare to other structure prediction algorithms that utilizes minimal free energies?
6. How do the authors know that inline-seq is accurate. Can they compare their sequencing results to traditional inline probing using capillary sequencing or sequencing gels?
7. After incorporating the different design rules, the increase in translation of their target RNA is 1.7X. Can the authors speculate what would be the additional designs to further push this increase even more? Can the authors dissect the relative contributions of each factor to the successful design of the RNA?
8. With the modifications and design, what does that mean with regards to the stability of the RNA at higher temperatures?

Reviewer #3:

Remarks to the Author:

A manuscript from Leppek et al. describes a holistic approach for sequence optimization of mRNA therapeutics using a suite of cellular and biochemical assays. It is timely, important, and impressive, although still requires some work to bring it to publishable form.

My major critique is with regards to the vast content of that paper. I see a tension between amount of detail and showing an end-to-end process of sequence optimization. I would have found a presentation of the work more compelling if it was split in two or even three independent manuscripts. That would allow clearer presentation of many of the important insights. On the other hand I see that the authors wanted to emphasize their end-to-end approach, and there may exist readers that would appreciate that.

Comments:

- (p4 and beyond) the pooled mRNA library is in vitro polyadenylate and capped. If efficiencies of those reactions are different for different mRNAs in the pool this may affect the results. Please

show/discuss that this is not the case (are both of the reactions ~100% efficient?)

- (p7) optimization of CDS sequence which results in shift of mRNA to heavier polysome fractions may be a result of increase in translation, or alternatively it may stem from slower translation (longer residency of ribosomes on mRNA with the same loading). This possibility was not discussed/assessed in the manuscript.

- (p7) you are writing that you "found 5' UTRs that are highly structured [...] can support efficient translation". However, you are not showing what is the 'average' impact of structured 5' UTRs on translation. Is dengue virus's 5'UTR a rare outlier?

- (Fig 2A, half-lives) - how good are those fits to the decay curves? Would it be more fitting to show horizontal lines of let say 95% conf. intervals instead of point estimates?

- (Fig 2B) in results you are writing that "constructs with higher ribosome load values tended to be more unstable". The effect shown on the first plot show correlation of -0.012, which is very small and unlikely to be significant. Is that statement supported then?

- (p10) you are explaining extensively (also with figures) that the cumulative protein production depends more on mRNA half-life than on ribosome load. Isn't it equivalent to simply stating that you observe more variation in mRNA in cell half-life than with ribosome load? when you model protein output using your data what % of variation is explained by which of those readouts?

- (p11) you are writing "the largest changes in in-solution stability occurred across CDS variants" - is it simply because CDS is the longest element or is variation in per-bond hydrolysis rate largest in the CDS part?

- (p13, top of the page) - is it possible that some of those triloops are actually larger in solution making them more labile?

- (p13, top) - what is the sequence preference of CirLigase used for that? Is it possible that it affected the observed sequence determinant of degradation?

- (p13) you are comparing performance of DegScore with other metrics, comparing Pearson's R's. Seeing the numbers it seems quite unlikely that DegScore is significantly better than 'summed unpaired probability of the RNA structure ensemble' score. Please do a statistical test if DegScore is superior to any of those other metrics. [see also comment to Fig 3E below]

- when discussing impact of pseudouridine on stability, please also consider/evaluate that it may be a result of changing RNA secondary structure (see e.g.

<https://www.ncbi.nlm.nih.gov/pmc/articles/PMC3950712/>)

- (p15) you are writing that 'LinearDesign-1 and GCrich_2 achieved similarly high protein levels compared to Nluc start. However, seeing the figure it looks like to latter should be '-423.7' ?

- (p18) you write: "biophysical and biophysical"

- (Fig 1E) to help reader please make a small separator between top, middle and bottom genes.

- (Fig 2C and E) the 'C' part only shows that the predictor puts more weight on half lives. Can there be a better way to show this? 'E' part: top - protein predicted after how long, 24h? bottom - confusing with changes order. You are writing 'note closer similarity' - please show Spearman's rho.

- (Fig 2D) - m0 and l are described in legend, but not used in formula

- (Fig 3A) on the drawing it shows "3' UMI" but from the description it suggests that it is a sample barcode not an molecular identifier?

- (Fig 3C) what does it show? is it average for this class of triloops? Not explained.

- (Fig 3D) y-axis?

- (Fig 3E) what are the stars? Yellowstone and LinearDesign? Not explained.

- (Fig 3E) use of Pearson's correlation is not appropriate, please use Spearman's.

- (Fig 4G) this begs to ask how well those curves can be predicted using data from panel C?

- (S.Fig 3A) y-axis?

Minor points:

- please make sure to include a link to GEO repository

- equation of a kinetic model is shown at least 3 times, and derivation explained twice (main text and supplement)

Reviewer #1 (Remarks to the Author):

In this study, Leppék et al reported an RNA sequencing-based platform to measure and predict in-cell mRNA stability, translation efficiency, and in-solution mRNA stability. The authors selected a group of synthetic mRNAs (a total of 223) with different combination of 5'UTR, CDS, and 3'UTR. The major goal is to establish some “rules” guiding mRNA design to achieve optimal stability and translation efficiency.

Therapeutic mRNA engineering is of significance given the recent success of mRNA vaccines targeting coronavirus. From RNA structures, ribosome loading, stability, and modification, the authors employed comprehensive approaches in order to establish “superfolder” mRNAs. Unfortunately, after finishing reading of the manuscript, this reader is totally lost. What exactly is the “superfolder” mRNAs that hold therapeutic potential?

We thank the reviewer for his/her comments and insights. We now more clearly define what we term “superfolder” mRNAs. In particular, we classify “superfolder” mRNAs to be those which are predicted to have highly structured CDSs. We now define this terminology in the beginning of the introduction so as to familiarize the reader upfront with the terminology.

More puzzling, some of the conclusions are contradictory to the conventional view. For instance, how could structured CDS improve in-cell translation efficiency?

Previous screens have been limited to short variable sequences, particularly in the UTRs and not in the CDS region, that typically do not compare mRNA stability and protein output in parallel. By carrying out a comprehensive screen to assess the contribution of ribosome load and mRNA stability of defined regions in a controlled setting and within the context of full-length mRNAs, we have found potentially new principles of RNA structure-function relationships and their effect on mRNA stability and translation. For example, highly structured CDSs improve in vitro mRNA stability while maintaining mRNA translation or slightly improving it (Fig. 4C).

The reviewer brings up an interesting point in regards to how a structured CDS can maintain high levels of translation. Our data indicate that enhanced mRNA stability is the main contributor to the end-point luciferase output we measure. That said, we are actively engaged in understanding the mechanism by which a highly structured CDS can still be well translated. One potential explanation for this could be that the ribosome has strong intrinsic helicase activity which can unwind stable CDS structure without detriment to the protein output (Takyar et al., 2005 doi: 10.1016/j.cell.2004.11.042). We have now further observed that our optimized CDS structured mRNAs can even support translation when complexed into lipid particles that are transfected into cells. For these results we have added a new Figure in the paper (**Fig 5**).

How could polysome-associated mRNAs are less stable?

With regards to polysome-association, we discuss that heavily ribosome-loaded mRNAs might actually be subject to ribosome collisions which triggers mRNA decay such that mRNAs are more stable with an optimal number of ribosomes loaded for efficient protein synthesis.

The entire manuscript also suffers from many overstatements. Overall, this manuscript does not seem to deliver the useful information as it is supposed to do.

We revised the manuscript text such that potential overstatements are more carefully stated and conclusions are defined more clearly throughout the text. We particularly use more specific language to express the description of an “optimized” parameter.

Main concerns

1. The entire library contains only 223 mRNAs, which is far from “the first large-scale screen of mRNA redesigned across its entire length”. Given the large variations in 5'UTR, CDS, and 3'UTR, how robust is the statistical power?

There has not been a screen that used the variation of complete regions (5' UTR, CDS, 3'UTR) in their entirety. To address the effects of each region, we only vary one region at a time or two in combination, keeping either hHBB 5' and 3' UTR sequences constant for Nluc CDS structure variants, or varying 5' or 3' UTRs and keeping the starting Nluc reporter CDS and hHBB 3' and 5' UTR constant, respectively. Then we also include 5' and 3' UTR comparisons to address combinatorial, synergistic effects. So far, mRNA screens for translation and stability have mostly only focused on small regions in the 5' UTR (Jia et al., 2020 doi: 10.1038/s41594-020-0465-x;) (Sample et al. 2019, doi: 10.1038/s41587-019-0164-5) or 3' UTR (Zhao et al., 2014, doi: 10.1038/nbt.2851; Orlandini von Niessen et al., 2019, doi: 10.1016/j.ymthe.2018.12.011), which we cite in the paper. There has not been a study that was able to analyze the contribution of complete portions of the mRNA because full-length and barcoded mRNA template synthesis for a pooled screen was not feasible or too expensive. We were only able to do these experiments because we could leverage the Codex DNA template synthesis platform, which has only recently become commercially available. Since we did not completely randomize mRNA regions but included literature-based and rationally designed mRNA sequences, this provided a library of 233 different reporter mRNAs. We now tone down the description of our screen in the introduction and results.

For all summary metrics that we have made across the variant library (in-solution degradation coefficient, in-cell degradation coefficient and average ribosome load), we provide error estimates based on replicate measurements and conservative bootstrapping methods (described in the methods) for each of the constructs in the library in **Suppl Table S1**.

2. Figure 1E, it is quite surprising to find that the commonly used beta-actin (hACTB) 5'UTR is the poorest in ribosome loading. By contrast, the scramble 5'UTR (scrUTR) behaved much better. The authors did not even discuss about this.

Indeed, the hACTB does not perform well in terms of ribosome loading but the mouse beta-actin (mActb) performs in the same range as the scrUTR. The scrUTR is a very short 5' UTR of only 46 nt that supports efficient translation (Osuna et al., 2017, doi: <https://doi.org/10.7554/eLife.27949>). This may be because the scrambled short 5' UTR has low GC content (48% GC) and more likely to be unstructured which favors scanning. In addition, in the translation field, more so than hACTB, hHBB is generally preferred as an example of an efficiently initiated 5' UTR. hHBB performs better in our ribosome load analysis than the ones mentioned here. We now add a sentence to refer to this finding.

3. Figure 1 shows GC-rich CDS has almost the highest ribosome loading. In human cells, it has been believed that increasing GC-content can improve protein output because GC-rich codons tend to be optimal for translation in human cells. Therefore, a lower ribosome density is expected in GC-rich CDS, since elongating ribosomes move faster on these mRNAs. This is quite opposite to ribosome loading reported in Figure 1.

While our “GC_rich” CDS designs are not specifically designed for codon optimization, they do inherently have some of the highest codon adaptation index (CAI) values of our panel (e.g. GC_rich_4 = 0.840 CAI), likely due to the bias of GC residues in optimal codons as the reviewer points out. In particular, the GC_rich designs were chosen based on their predicted ability to form extensive RNA secondary structure as shown in Figure S7A. As a result, we hypothesize that the increased RNA secondary structure may counteract the conferred benefit of high CAI in these designs. In turn, this may strike a balance between codon optimality and RNA structure such that the dwell time of ribosomes on these transcript remains longer, thereby increasing the measured ribosome load. Moreover, we would like to point out that the ribosome load measured in our dataset for the GC_rich

designs is not particularly high and is comparable to the non-optimized Nluc reference sequence (~1.6). We have further discussed this in the text and added a new Supplemental Figure S4 which includes the correlation of CAI and GC content with ribosome load and in-cell mRNA half-life.

4. Supplemental Figure 2. The massively paralleled assay based on the 35 random nucleotide sequence in 5'UTR is expected to generate some exciting “principles” in optimal mRNA design. Oddly, these results were described in a brief manner.

We set out to design a workflow for in-cell translation efficiency with the specific question whether we can further optimize sequence features upstream of the Kozak and AUG sequence. We on purpose chose hHBB 5' UTR as a model and asked if we can increase initiation and translation efficiency of a hybrid hHBB 5' UTR by randomizing the 3' most 35 nt to the Kozak.

We added a separate page of result description in the “Supplemental Text” to describe this series of experiments in greater detail due to length restrictions of the main text. Moreover, we chose not to combine this experiment with our full-length UTR/CDS variant data in the main text as it is an experiment of “5' UTR sequence selection” over several rounds of selection by polysome fractionation rather than a pooled screen and one endpoint measurement as in Figure 1. Nevertheless, we learn several design principles such as a clear de-
richment of out-of-frame start codons and favored stem-loops in the middle of the 5' UTR. We describe these main findings in a full paragraph in the main text and refer the reader to the Supplemental Text.

5. Figure 2 shows that mRNAs with higher polysome/monosome ratio tends to have shorter half-life. Does that suggest translating ribosome trigger RNA degradation? Without considering other factors, e.g. sequence length or different sequence composition (e.g. different GC-content), it is not convincing to make a conclusion that polysome mRNAs are short half-life mRNAs. In fact, from the right panel of Figure 2B, only a small subset points (mRNAs) shows strongly negative correlation of half-life to polysome/monosome. They may use their random sequence reporter (Supplemental Figure 2) to further investigate such correlation.

The negative correlation of high polysome/monosome ratio with low mRNA stability is mostly driven by 5' UTR variants, strongly suggesting that the underlying effect has to do with ribosome load and not CDS length or composition (which are constant in those 5' UTR variants). We want to point out that specifically for the CDS variants of the same CDS length but different RNA structure, we see a wide range of mRNA half-lives with similar polysome/monosome ratios. We now added a new Supplemental Figure S4 which includes the correlation of CAI and GC content with ribosome load and in-cell mRNA half-life.

Our observations indeed suggest that the translating ribosome triggers mRNA degradation. We mention in the discussion the possibility of mRNA decay induced by increased ribosome collisions upon heavy loading of ribosomes and cite other references that have discussed this possibility. This would indicate that an expression optimum requires a balance of ribosome loading and mRNA stability. We now add the individual correlations of ribosome load and in-cell mRNA half-life with protein expression at 6h and 12h (Suppl. Figure S4B). This analysis indicates that mRNA half-life is only an approximate predictor of protein expression and that it predicts expression better at later time points than early timepoints, which is expected from the model that includes both ribosome loading and mRNA stability.

Regarding the random sequence reporter, the in-cell translation selection experiment is unfortunately not helpful here as in this experiment we only select for the heaviest loaded mRNAs and select that pool further with again a focus on the most heavily loaded mRNAs, not monosome fractions. This does not allow us to interpret relative distributions between monosome/light polysomes/heavy polysomes.

6. Figure 3F, it is unclear how the authors conclude that “substitution of U with ψ or m1 ψ led to remarkable suppression of in-line hydrolysis”. Is the 1.2 fold increase (Figure 3G and 3H) statistically significant?

We have updated our introduction of Figure 3F to better explain that the constructs depicted in Figure 3F are representative structures from more extensive data: “We observed that substitution of U with either ψ or $m^1\psi$ led to suppression of in-line hydrolysis at the substituted residues, presumably through changed nucleophilicity at the site of substitution (example constructs in Fig. 3F, statistics over all constructs probed by capillary electrophoresis in Fig. S5)”.

We now include two-sided pairwise significance tests on the fit half-lives in Figure 3H, and do find that for all three constructs shown, the difference between the half-lives with and without pseudouridine have significance $p < 0.0001$. We reference this in the manuscript on page X: “In-solution mRNA half-lives were measured using capillary electrophoresis to evaluate the fraction of intact mRNA over time (Fig. 3G-H, pairwise significance comparisons between constructs in Fig. 3H inset).” We note that the half-life fits and decay curves currently included in Fig. 3H differ from the panel originally presented, although the underlying electrophoresis data (shown in Fig. 3G) are the same and visually show striking differences in half life. This was due to an inconsistency in the fitting code used for Fig. 3H compared to all other half-life analyses in the manuscript, which we updated. We also now provide significance analysis confirming the stated half-life differences, displayed in the inset bar graph of Fig. 3H.

Fig. 3H reproduced below:

7. Figure 4A, the translational output assay is probably the most useful in judging the mRNA potency. However, the luciferase activity (especially from 24 hr) is largely inconsistent with the previous design “principles”.

We are unsure of the inconsistencies the reviewer is referring to. The polysome-seq and ribosome load experiment in Figure 1 was based on 6h expression of the mRNA pool after transfection and thus is more in line with the 6h luciferase assay data in Figure 4A. Importantly, ribosome load as a measure of the distribution of an mRNA in a sucrose gradient according to weight as a proxy for ribosomes loaded does not directly infer nor is expected to directly correlate with protein output of an mRNA, which also depends on in-cell half-life. We added text to the main text to highlight that.

The assay in Figure 4A was meant to combine various UTRs and CDS designs, ones that had performed well in terms of ribosome load and in-cell mRNA stability in the screen, in an effort to identify an overall improved mRNA design. Consistent with our finding that mRNA half-lives are the main driver of overall protein output (Figure 2), the LinearDesign CDS combined with Hbb 5’/3’ UTRs express the most luciferase by 24h. While we find the CoV-2-UUG-dSL-3/DEN2 maintains good expression of luciferase at 6h consistent with a higher ribosome load, this expression is lost by 24h which is agreement with the shorter mRNA half-life exhibited by these UTRs (Figure 2). As a result, we based our following experiments on these findings and chose to proceed with hHBB 5’ and 3’UTRs and further improve CDS design.

8. For the therapeutic point of view, how to achieve “CDS” design? The coding region does not have lots of room for engineering, except changing codon optimality. On the other words, how to create “superfolder” mRNAs by changing CDS sequences? This is utterly confusing.

We aim to better contextualize the existing literature from bioinformatics on stabilizing mRNA sequences with the following added text on page 17: “A simple calculation of the number of synonymous mRNAs for the Nluc CDS (621 nt) using a human codon table reveals that there are 1.6×10^{101} sequences possible. The hypothesis that mRNA stability may be increased by changing the CDS sequence has been explored theoretically through algorithms designed to optimize the predicted minimum free energy of a coding sequence (Cohen & Skiena, 2003 doi: 10.1089/10665270360688101; Terai et al., 2016 doi: 10.1093/bioinformatics/btv678; Zhang et al., 2020 arXiv:2004.10177), as well a biophysical model for hydrolysis that predicted a minimum two-fold decrease in hydrolysis could be achievable (Wayment-Steele et al., 2020 doi: 10.1101/2020.08.22.262931), but these predictions for increases in stability have not been experimentally validated.” The current work validates these predictions experimentally by using several of the aforementioned algorithms, as well as the Eterna platform for rational RNA design, to achieve highly structured “superfolder” mRNAs for the short Nluc and longer EGFP sequence. We also state more clearly in the text that a high degree of predicted structural diversity can be observed among superfolder mRNAs (page 17): “These 24 different CDS designs displayed a wide variety of structural diversity even for a short ORF such as Nluc (see **Fig. S7**).”

Reviewer #2 (Remarks to the Author):

The manuscript by Leppek et al uses different strategies to optimize the translatability and stability of mRNAs inside cells and in solution in order for maximum protein production for RNA vaccines. They developed a strategy to test ribosome load and decay rates of their RNAs and tested different 5' and 3'UTRs as well as different CDS features using cloud sourcing. They observed that decay is more important than ribosome load for long term translation. They also identified structural features in the CDS and showed that replacing U with pseudouridine increases protein production. Collectively, they were able to improve the protein production of their target RNAs using their design rules. This paper comes at an important time whereby RNA vaccines show great promise in our combat against SARS-CoV-2. However, current versions of RNA vaccines still suffer from degradation issues and would benefit from better design in terms of stability and translatability. I have some major concerns that needs to be addressed before this manuscript should be published.

1. The authors use reverse transcription, PCR and sequencing to identify full length transcripts that have not undegone in cell or in solution decay. They conclude that the lengths of CDS make a difference to stability. How do the authors know that the differences in decay is not due to increased RT stoppages (dropoff) along the transcript because of longer or more structured CDS/UTRs?

We thank the reviewer for their comments. We were aware of the concern of the reviewer with regard to potential RT dropoff or biased preference of RT readthrough dependent on CDS length (EGFP and Nluc are very different in length) or structure which could be a challenge for the RT to generate full-length products.

However, we are able to resolve this issue because we are quantifying relative differences across timepoints, from which we fit decay curves and half-lives. Thus, any construct-specific biases in RT efficiencies are regressed out (i.e., read counts are normalized by average abundances of each construct; and only the relative ratios of the counts between timepoints, not the absolute read counts, are used for interpreting stability). Thus even if an mRNA has a high rate of increased RT stoppage, that reduction in final cDNA will be reflected as a constant suppression of the number of reads in every time point and will not impact the final fitted half-life. We clarify this now more in the main text and also now give examples of fitted decay curves for all 233x constructs and their respective relative distributions in the sucrose gradient (**Fig S1E**) as well as for their relative differential mRNA decay over 4 time points in the in-cell mRNA half-life assay (**Fig S1F**) based on this readout for individual constructs.

2. From their screens, Nluc with 5' and 3'den2 UTRs have high ribosome load. Why was the truncation and optimization done on SARS-CoV-2 UTRs and not den2 5'and 3'UTRs? In addition, den2 5'UTR data is not illustrated in Figure 2E.

This is a great point. We indeed see that the combined DEN2 3' and 5' UTRs are well loaded with ribosomes (**Fig 1E**). Indeed, we included the DEN2 UTRs because it has been shown before that the 3' and 5' UTR structures in the dengue virus RNA genome are interacting for improved translation of the viral genome (Chiu et al., 2005, doi: 10.1128/JVI.79.13.8303-8315.2005; Holden & Harris, 2004, doi: <https://doi.org/10.1016/j.virol.2004.08.004>) . However, the DEN2 5' UTR alone did not perform as one of the top candidate 5' UTRs.

We chose to perform mutagenesis on the SARS-CoV2 5'UTR because it contains significantly more structured regions than the DEN2 5' UTR and has been shown to support sustained translation despite its high RNA structure content. In particular, we hypothesized that changes in the structure and/or sequence would lead to an increased ribosome load while maintaining the benefits of a highly structured RNA on in-cell and in-solution stability.

We note that we did further explore DEN2 sequences later in the study. Due to its positive effect on ribosome load, we choose the DEN2 3' UTR, together with the best SARS-CoV2 5' UTR variant, as one of the

overall best performing 3' UTRs for the combinatorial 5'/3' UTR analysis on protein expression in Fig. 4A. We now clarify these choices through additional sentences in the manuscript.

3. What does a higher ribosome load of 1.7-2.3 mean? How many fold increase in translation does that correspond to?

The metric of ribosome load (see Fig. 1D) takes the number of ribosomes per fraction and the percent mRNA per fraction into account. We use it as an estimate for how many ribosomes are loaded per mRNA construct (average ribosome number per mRNA), and have assumed that this is proportional to the translation rate per mRNA per unit time. With these assumptions, a high ribosome load of 2.3 compared to a reference of 1.6 (for our reference Nluc CDS with hHBB UTRs) would correspond to a 1.4x increase in translation rate. We now clarify these points through text changes in the manuscript.

4. With regards to the library of randomized 5'UTR sequences, how long are the 5'UTRs and how complex is the library?

The 5' UTRs in the selection experiment have a fixed size of total 70 nt: 29 nt of the 5' most part of the hHBB 5' UTR, followed by a 35 nt randomized region (N35), and then the 6 nt Kozak consensus sequence. The initial starting complexity of each degenerate pool is $\sim 2.4 \times 10^{11}$. Given the practical limitations in sequencing depth and to account for sequencing errors, we limited our analysis to ~ 670000 final clusters of unique sequences above a minimum read count (≥ 10). We have now included these details in the methods section describing the selection experiments.

5. How does the results of lineardesign mRNA structure optimization compare to other structure prediction algorithms that utilizes minimal free energies?

There are two other free-energy-guided design algorithms in the literature (Skiena and Cohen) and CDSfold -- neither were available to us at the time of the study. We have added text to contextualize these algorithms on page 17: "The hypothesis that mRNA stability may be increased by changing the CDS sequence has been explored theoretically through algorithms designed to optimize the predicted minimum free energy of a coding sequence (Cohen & Skiena, 2003 doi: 10.1089/10665270360688101; Terai et al., 2016 doi: 10.1093/bioinformatics/btv678; Zhang et al., 2020 arXiv:2004.10177)". We have recently encouraged the CDSfold authors to release their code, and look forward to comparing the resulting designs to LinearDesign in future studies.

6. How do the authors know that inline-seq is accurate. Can they compare their sequencing results to traditional inline probing using capillary sequencing or sequencing gels?

We now include representative profiles of 4 RNAs comparing In-line-seq degradation profiles to conventional capillary electrophoresis-measured degradation in the supplement (**Suppl. Fig. 5, reproduced below**). We reference this figure in the main text in the Results, "We compared degradation profiles from In-line-seq to profiles from constructs whose in-line degradation was probed one-by-one and read out with capillary electrophoresis, which confirmed excellent agreement between In-line-seq and low-throughput capillary electrophoresis (**Fig. S5**)".

7. After incorporating the different design rules, the increase in translation of their target RNA is 1.7X. Can the authors speculate what would be the additional designs to further push this increase even more? Can the authors dissect the relative contributions of each factor to the successful design of the RNA?

We want to highlight that we achieve an improvement of translation by 1.7-fold after 24h of expression of the 'winning' construct compared to the start sequence. This optimized construct (a combination of LinearDesign, RiboTree, and DegScore algorithms) is able to maintain or slightly improve protein output *despite* highly increased CDS structure (Fig. 4C). However, in addition to increased translation, we see for our top mRNA candidates an improvement of their in solution stability of up to 2.5x, which qualifies them to be the overall best performing mRNA designs. In a new figure in the paper (Fig. 5), we have observed that even when complexed with lipid particles, our optimized RNA designs show improved translation efficiency inside cells after transfection. Finally, it is worth noting that all mRNAs tested in Fig. 4C had pseudouridine incorporated, thus the actual increase compared to our reference sequence without pseudouridine is likely much higher.

In the future, we speculate that by performing inline-seq and PERSIST-seq on thousands of long mRNA sequences, this would enable the discovery of sequences and structures that enhance RNA stability in solution and in cells even further. Such data could further enhance our DegScore predictor and allow development of an in-cell half-life predictor; both could guide future CDS structural designs. At the moment, it is difficult to directly assess the contribution of each element in the successful RNA designs given the confounding variable that each element presents, and we have tried to avoid speculation in the manuscript itself. That said, our results suggest that a given 5' or 3' UTR has a greater influence on ribosome load than stability (Fig 1, 2), while inclusion of a

structured CDS greatly influences the stability of the mRNA both in-cell and in solution. We think it is the combination of both of these that allowed for the design of a highly translated and stable mRNA.

8. With the modifications and design, what does that mean with regards to the stability of the RNA at higher temperatures?

We share the reviewer's interest in the effect of higher temperature on RNA stability, and have included further analysis of our In-line-seq experiments performed at 50°C to more explicitly compare degradation rates due to increased temperature compared to degradation caused by Mg^{2+} and increased pH. We calculated the total degradation rates from the short RNA fragments in the four In-line-seq experiments and found that the degradation rates from the two conditions with increased temperature are highly correlated to degradation with increased pH. We have included this analysis in Figure S4 and address this in the main text on page 13: "We matched the accelerated degradation conditions used for PERSIST-seq in-solution stability measurements, but also verified that calculated degradation rates without Mg^{2+} , at lower pH, and higher temperature gave strongly correlated results (Fig. S5A)."

Fig S5A reproduced below:

Reviewer #3 (Remarks to the Author):

A manuscript from Leppek et al. describes a holistic approach for sequence optimization of mRNA therapeutics using a suite of cellular and biochemical assays. It is timely, important, and impressive, although still requires some work to bring it to publishable form.

My major critique is with regards to the vast content of that paper. I see a tension between amount of detail and showing an end-to-end process of sequence optimization. I would have found a presentation of the work more compelling if it was split in two or even three independent manuscripts. That would allow clearer presentation of many of the important insights. On the other hand I see that the authors wanted to emphasize their end-to-end approach, and there may exist readers that would appreciate that.

Comments:

- (p4 and beyond) the pooled mRNA library is in vitro polyadenylated and capped. If efficiencies of those reactions are different for different mRNAs in the pool this may affect the results. Please show/discuss that this is not the case (are both of the reactions ~100% efficient?)

We thank the reviewer for the constructive comments. The different mRNA constructs are designed such that the most 5' and 3' ends are identical (see Suppl. Fig. 1A). We on purpose added these constant regions through which the whole library is amplified in one pot. All other steps including capping and polyadenylation are occurring in the same reaction also. We have no indication that for such RNA modifications in a pool there would be a bias for certain sequences or structures in the different mRNA. By denaturing agarose-gel electrophoresis before and after cap and polyA addition (**Suppl. Fig. 1C**) we see an upwards shift of the whole library by extension of all mRNAs by 150 As. However, it is very hard to assess the exact efficiency of such RNA modification and we now added this notion to the results on page 4: "We have no indication for any bias in the efficiency of such RNA modifications for certain sequences or structures in the different mRNA when performed in a pool (**Fig. S1C**)."

In addition, for confirmation of expression by individual RNA designs, we performed PCR amplification, IVT, and capping and polyA-tailing individually (Fig. 4A), for which we did not see a construct-dependent effect on either of these steps when we compare 18 different constructs of different CDS and 5'/3' UTR sequence. We now document the whole RNA synthesis workflow in a new **Suppl. Fig. 9**.

- (p7) optimization of CDS sequence which results in shift of mRNA to heavier polysome fractions may be a result of increase in translation, or alternatively it may stem from slower translation (longer residency of ribosomes on mRNA with the same loading). This possibility was not discussed/assessed in the manuscript.

The reviewer is correct that it is possible that an increase in CDS structure might slow down elongating ribosomes as they unwind more extensively double-stranded mRNA regions, while having the same translation initiation rate as less structured CDS mRNAs on the identical hHBB 5' UTR. We have added a statement to the text discussing these different scenarios.

- (p7) you are writing that you "found 5' UTRs that are highly structured [...] can support efficient translation". However, you are not showing what is the 'average' impact of structured 5' UTRs on translation. Is dengue virus's 5'UTR a rare outlier?

We think that our literature-based approach of finding structured 5' UTRs from viral genomes or, more generally, eukaryotic UTRs that favor translation and stability, was a choice to generate a catalog of existing sequences or structures from all kingdoms of life that would be useful for RNA therapeutics designs. Thus, we did not gradually increase RNA "structuredness" in the 5' UTR to be able to make a claim for the average impact of 5' UTR structures but rather compare diverse natural 5' UTRs. It would be difficult to delineate gradual increases of

structuredness in 5'UTR elements that exist in nature. The Dengue virus as well as several viral 5'UTRs that we have included are relevant examples of how overall structured 5'UTRs can support translation.

- (Fig 2A, half-lives) - how good are those fits to the decay curves? Would it be more fitting to show horizontal lines of let say 95% conf. intervals instead of point estimates?

We provide the error estimates for the fits of degradation coefficients in the supplemental table (**Table S1**). Furthermore, we also now give a full plot of individual replicate abundance measurements from each timepoint and across all constructs, with fitted decay curves overlaid on top (**Fig S1F**). This should both better illustrate how the half-life values are derived for the readers as well as provide an idea how good the fits are. Additionally, a similar plot is now also provided for ribosome load and relative mRNA distributions across the sucrose gradient experiment (**Fig S1E**).

- (Fig 2B) in results you are writing that “constructs with higher ribosome load values tended to be more unstable”. The effect shown on the first plot show correlation of -0.012, which is very small and unlikely to be significant. Is that statement supported then?

We did not intend this sentence to mean that there is correlation or a linear relationship between ribosome load and stability. Rather, we meant the opposite - we were pointing out that while the correlation between ribosome load values and stability is close to 0 ($r_s = -0.012$, $p = 0.82$), moderate increases in ribosome load (below ~ 1.7 for Nluc constructs) seems to be positively associated with mRNA stability while ribosome load values higher than this range is associated with low mRNA stability.

Examining this non-linear relationship more closely, we find that metrics which separately look at polysome versus monosome loading better capture this effect: high polysome/monosome ratio predicts lower stability ($r_s = -0.516$, $p < 2.2e-16$) while looking at monosome/pre-80S ratios predict increased stability ($r_s = 0.467$, $p < 2.2e-16$). One clarifying detail is that many of our 5'UTR variants led to strong polysome loading (over monosomes), while the more moderate dynamic range of ribosome load differences seen in CDS variants are mainly due to differences in monosome loading (over pre-80S). We have now added Supplemental Figure S3, which plots Figure 2B separately for each group and this should better illustrate our observations about the highest ribosome loading (via increasing in polysomes) leading to unstable mRNAs.

- (p10) you are explaining extensively (also with figures) that the cumulative protein production depends more on mRNA half-life than on ribosome load. Isn't it equivalent to simply stating that you observe more variation in mRNA in cell half-life than with ribosome load? when you model protein output using your data what % of variation is explained by which of those readouts?

While it is unclear what a good summary metric for our model may be (which would be analogous to r^2 for linear models), in order to make this point straightforward, we have added Supplemental **Fig S4B** which shows individual correlations of mRNA half-life or ribosome load with protein expression. This should more clearly illustrate the point about mRNA half-life being a better predictor of total protein expression at later timepoints (also related to comment about **Fig. 2C and E** below). In a new figure in the paper (**Fig. 5**), we have observed that even when complexed with lipid particles, our optimized RNA designs show improved translation efficiency inside cells after transfection.

- (p11) you are writing “the largest changes in in-solution stability occurred across CDS variants” - is it simply because CDS is the longest element or is variation in per-bond hydrolysis rate largest in the CDS part?

The large change in in-solution stability with respect to CDS changes is due to the large range in lengths as well as 'structuredness' that was intentionally designed into those variants, as well as use of primer pairs that primarily captured mRNA windows in which the CDS was the longest element. Rather than call these effects the 'largest changes', we now simply state that the range of half-lives was greater than 3-fold.

- (p13, top of the page) - is it possible that some of those triloops are actually larger in solution making them more labile?

To address this possibility, we have updated **Fig 3C** and the corresponding panels in **Fig S4** to also include the base stack at the base of the triloop. In SHAPE reactivity data in **Fig S4C**, we observe that the Watson-crick stack below the triloop has low reactivity, indicating that the nucleotides below the triloops are paired in the data analyzed.

- (p13, top) - what is the sequence preference of CircLigase used for that? Is it possible that it affected the observed sequence determinant of degradation?

To address the possibility of ligation biases from CircLigase, we compared SHAPE reactivity and in-line degradation of 4 constructs using capillary electrophoresis to the results obtained in high throughput, which is now included in **Fig S5**. We observe excellent agreement between the low-throughput in-line degradation and In-line-seq, suggesting that sequence preferences from CircLigase do not affect the stated results from In-line-seq. We reference these results in the main text on p. 13: "We compared degradation profiles from In-line-seq to profiles from constructs whose in-line degradation was probed one-by-one and read out with capillary electrophoresis, which confirmed excellent agreement between In-line-seq and low-throughput capillary electrophoresis (**Fig. S5**)."

- (p13) you are comparing performance of DegScore with other metrics, comparing Pearson's R's. Seeing the numbers it seems quite unlikely that DegScore is significantly better than 'summed unpaired probability of the RNA structure ensemble' score. Please do a statistical test if DegScore is superior to any of those other metrics. [see also comment to Fig 3E below]

We have performed the requested significance test (with updated Spearman correlation) and found the results are nonsignificant. We describe this test and finding in the main text on page 14: "The difference in correlation coefficients were not significant with $p < 0.05$ as evaluated by a two-sided significance test on dependent overlapping correlations (see Methods); more experimental studies are likely needed to achieve significance for these and future metrics."

We have also included further analysis modifying the DegScore model to account for the stabilizing effect of pseudouridine (PSU), and found that this resulted in improved correlation to the final 24 Nanoluciferase constructs in Fig. 4E. We describe this in Methods, "For the 24 Nanoluciferase constructs synthesized with pseudouridine (PSU), we wished to test if accounting for increased stability of PSU resulted in improved performance. Without accounting for PSU, we found that the DegScore predictions had a correlation of (Spearman $R = 0.55$, $p < 0.001$) for the 24 constructs. We modified DegScore to set the prediction per nucleotide at each Uridine to be zero, to mimic minimal contributions to degradation. This simple change to the DegScore model resulted in a notable improvement in correlation (Spearman $R = 0.67$, $p < 0.001$, Figure 4E), and was used in analysis for constructs synthesized with PSU."

- when discussing impact of pseudouridine on stability, please also consider/evaluate that it may be a result of changing RNA secondary structure (see e.g. <https://www.ncbi.nlm.nih.gov/pmc/articles/PMC3950712/>)

We now cite the mentioned reference while discussing the possibility of changes in secondary structure due to pseudouridine. We also cite Mauger et al. 2019 (doi: 10.1073/pnas.1908052116), who measure optical melting of mRNAs synthesized with unmodified and pseudouridine and find a small effect of PSU stabilization, roughly 0.25 kcal/mol per base pair, indicating that the effect of PSU on secondary structure is stabilizing (though weakly so). On page 15, "Structure mapping data by SHAPE and DMS profiling confirmed that ψ and $m^1\psi$ substitutions did not change the chemical reactivity of the RNAs outside the substituted positions, consistent with no change in global secondary structure; the suppression of in-line hydrolysis appears to be due to local chemical or

structural effects. However, more detailed measurements are needed to more holistically understand the effect of pseudouridine on secondary structure.”

- (p15) you are writing that ‘LinearDesign-1 and GCrich_2 achieved similarly high protein levels compared to Nluc start. However, seeing the figure it looks like to latter should be ‘-423.7’ ?

That is correct and we corrected this in the text.

- (p18) you write: “biophysical and biophysical”

Thank you for catching that. We corrected that in the text.

- (Fig 1E) to help reader please make a small separator between top, middle and bottom genes.

We now included a line to separate and name the top/middle/bottom categories of selected constructs shown in **Fig. 1E**.

- (Fig 2C and E) the ‘C’ part only shows that the predictor puts more weight on half lives. Can there be a better way to show this? ‘E’ part: top - protein predicted after how long, 24h? bottom - confusing with changes order. You are writing ‘note closer similarity’ - please show Spearman’s rho.

We have removed the length normalized panel to avoid confusion and added **Suppl Figure 4B** which shows individual correlations of mRNA half-life or ribosome load with protein expression to more clearly illustrate the point about mRNA half-life being a better predictor of total protein expression at later timepoints.

- (Fig 2D) - m0 and I are described in legend, but not used in formula

We fixed the legend and made sure it is consistent with the description in the text.

- (Fig 3A) on the drawing it shows “3’ UMI” but from the description it suggests that it is a sample barcode not an molecular identifier?

We have edited the figure label from 3’ UMI to 3’ barcode to have consistent terminology with the text.

- (Fig 3C) what does it show? is it average for this class of triloops? Not explained.

We have now explained more precisely in the text what this visualization depicts: “We visualized sequence dependence within triloops by collecting all triloops by sequence position at each position in the triloop. Each visualization in **Fig. 3C** represents the median degradation measured ($n > 5$).”

- (Fig 3D) y-axis?

A y-axis label of “Coefficient weight” has been added.

- (Fig 3E) what are the stars? Yellowstone and LinearDesign? Not explained.

We have added a legend to **Fig. 3E** to denote: these correspond to the same mRNA constructs Yellowstone (orange) and LinearDesign-1 (purple) which is consistent with the annotation in **Fig. 2G-I** as well as **Fig. 3G,H**.

- (Fig 3E) use of Pearson’s correlation is not appropriate, please use Spearman’s.

We agree and have replaced the main correlation coefficient shown in the text and **Fig 3E** with a Spearman correlation coefficient.

- (Fig 4G) this begs to ask how well those curves can be predicted using data from panel C?

Since x-axis is degradation time and y is percent degradation, we would indeed expect to see exponential curves with the same relative half-lives as shown in panel C (the difference in panel G from C is that G shows protein

expression in cells from transfected mRNAs following in-vitro degradation, whereas C shows mRNA levels). To clarify, we now show half-lives determined from the exponential decay fits to the data in G as a legend.

- (S.Fig 3A) y-axis?

Fig S3A y-axis label of "Reactivity (norm.)" has been added.

Minor points:

- please make sure to include a link to GEO repository

We now include the link of the deposited data in GEO, accession number GSE173083.

- equation of a kinetic model is shown at least 3 times, and derivation explained twice (main text and supplement)

We now made sure to not include duplications of the models.

Reviewers' Comments:

Reviewer #1:

Remarks to the Author:

In this revised manuscript, the authors addressed some of my previous concerns by clarification and additional analysis. The manuscript contains lots of data and it is challenging to present them in a coherent manner. The revised manuscript has been improved and I appreciate many novel points in terms of sequence optimization for therapeutic mRNA design. Since we are witnessing the beginning of the mRNA therapeutics revolution, this study will likely trigger many follow up application. Therefore, I think the revised manuscript has the merit to be published.

Reviewer #2:

Remarks to the Author:

The authors have answered most of my concerns, although I do agree with the other reviewer that differential polyA tailing efficiencies in different RNAs in the sample could affect their results. The experiment would benefit from having standardized, fixed length polyAs for all RNAs.

Reviewer #3:

Remarks to the Author:

Thank you for providing exhaustive answers to my review. I don't have any further comments and I consider the article to be ready for publication.